

# Diurnal Variability of Global Precipitation: Insights from Hourly Satellite and Reanalysis Datasets

Rajani Kumar Pradhan[1], Yannis Markonis[1], Francesco Marra[2,5], Efthymios I. Nikolopoulos[3], Simon Michael Papalexiou[1,4], and Vincenzo Levizzani[5]

[1]Faculty of Environmental Sciences, Czech University of Life Sciences Prague, Kamýcká 129, Praha – Suchdol, Czech Republic
[2]Departmentof Geosciences, University of Padua, Padua, Italy
[3]Department of Civil and Environmental Engineering, Rutgers University, Piscataway, NJ, 08854, USA
[4]Department of Civil Engineering, University of Calgary, Calgary, AB, Canada
[5]Institute of Atmospheric Sciences and Climate, National Research Council of Italy, Bologna, Italy

**Correspondence:** Rajani Kumar Pradhan (pradhan@fzp.czu.cz)

**Abstract.** Accurate estimation of precipitation at the global scale is of utmost importance. Even though satellite and reanalysis products are capable of providing high spatial-temporal resolution estimations at the global level, they are associated with significant uncertainties that vary with regional characteristics and scales. The uncertainties among precipitation estimates, in general, are much higher at the sub-daily scale compared to daily, monthly and annual scales. Therefore, evaluating these sub-

daily estimations is of specific importance. In this context, this study explores the diurnal cycle of precipitation using all the currently available space-borne and reanalysis-based precipitation products with at least hourly resolution at the quasi-global scale (60°N - 60°S), i.e., Integrated Multi-satellitE Retrievals for GPM (IMERG), Global Satellite Mapping of Precipitation (GSMaP), Climate Prediction Center Morphing (CMORPH), Precipitation Estimation from Remotely Sensed Information Using Artificial Neural Networks (PERSIANN), ECMWF Reanalysis v5 (ERA5). The diurnal variability of precipitation is

estimated using three parameters, namely, precipitation amount, frequency, and intensity, all remapped at a common resolution of 0.25° and 1 h. All the estimates well represent the spatio-temporal variation across the globe. Nevertheless, considerable uncertainties exist in the estimates regarding the peak precipitation hour, as well as the diurnal mean precipitation amount, frequency, and intensity. In terms of diurnal mean precipitation, PERSIANN shows the lowest estimates compared to the other datasets, with the largest difference observed over the ocean rather than over land. As for diurnal frequency, ERA5 exhibits

the highest disparity among the estimates, with a frequency twice as high as that of the other estimates. Furthermore, ERA5 shows an early diurnal peak and highest variability compared to the other datasets. Among the satellite estimates, IMERG, GSMaP, and CMORPH exhibit a similar pattern with a late afternoon peak over land and an early morning peak over the ocean. Overall, it emphasizes the need to integrate diverse datasets and exercise caution when relying solely on individual precipitation products to ensure a thorough understanding and precise analysis of global precipitation patterns.



## 1 Introduction

Precipitation is probably one of the most complex variables to estimate, given its high spatiotemporal variability in both space and time. This variability occurs in a continuum of timescales, ranging from inter-annual to seasonal and sub-seasonal (Hayden et al., 2023). In addition, the sub-daily scale variation, although on a longer scale driven by solar radiation, is highly regional (Yang and Smith, 2006). Thus, the accurate estimation of precipitation at the global level, and especially over the global ocean, is an ongoing challenge. The diurnal variation in global precipitation is of particular interest, considering its role in the global water cycle and climate system. Given the limitations of global climate models in accurately representing the diurnal cycle of precipitation (Tao et al., 2024; Trenberth et al., 2017), it is essential to have a robust reference to understand the processes responsible for this and to improve them. Satellite precipitation data offer a promising alternative that can serve this purpose. Therefore, understanding the precipitation characteristics across a range of scales from sub-daily (diurnal) to inter-annual variation at the global level not only improves our understanding of the water cycle and climate system but also helps the climate models to better represent small-scale processes (Marzuki et al., 2021).

Global satellite precipitation products provide a unique advantage for the estimation of the diurnal cycle, given their fine spatial-temporal resolution, near-global coverage, and access to remote areas (Levizzani et al., 2020a, b). Therefore, high-resolution multi-satellite estimates such as the Tropical Rainfall Measuring Mission (TRMM) (Yang and Smith, 2006), and the IMERG (Huffman et al., 2015) have extensively been used in the estimation of the diurnal variability at both regional (e.g. Hayden et al., 2023; Tan et al., 2019; O and Kirstetter, 2018; Dezfuli et al., 2017) and global (e.g. Watters et al., 2021; Watters and Battaglia, 2019) scales. For instance, Tan et al. (2019) recently evaluated the IMERG versions in terms of diurnal variation over the different regions of the globe. They found that IMERG V06 exhibits an improved ability to capture the diurnal variability than V05. In addition, it has also been reported that IMERG well represents the regional diurnal variability in a variety of climatic regions, such as over Africa (Dezfuli et al., 2017), Brazil (Afonso et al., 2020), China (Li et al., 2018) and the contiguous United States (CONUS) (O and Kirstetter, 2018). Furthermore, O and Kirstetter (2018) revealed the potential of IMERG as an alternative to ground measures over CONUS, even at a sub-daily scale. However, they also highlighted the fact that there are some region-specific biases to be considered. At the global scale, IMERG has shown promising results in capturing the key features of the diurnal cycle (Watters and Battaglia, 2019), and has been utilized as a reference to evaluate the ability of reanalysis and model datasets to represent diurnal variability (Watters et al., 2021).

Similarly, a substantial number of regional studies have inter-compared and evaluated various other satellite products in terms of sub-daily scale at different regions of the world (e.g., Roca et al. 2021; Pfeifroth et al. 2016; Sapiano and Arkin 2009; Janowiak et al. 2005). For instance, Afonso et al. (2020), evaluated the diurnal cycle of satellite estimates (IMERG, GSMaP, and CMORPH) over Brazil, finding improved performance in regions with deep clouds from local thermal heating compared to those dominated by shallow clouds. In addition, Ramadhan et al. (2023) noted that GSMaP products effectively capture the diurnal cycle of precipitation over the Indonesian Maritime Continent, although significant differences emerged in regions with more than one peak. Shawky et al. (2019) examined sub-daily (3h, 6h, and 18h) GSMaP and IMERG precipitation estimates in the arid region of Oman, concluding that GSMaP outperformed IMERG overall, despite issues identified with light and





extreme precipitation events. Recently, the GSMaP products have also been evaluated at an hourly scale by Lv et al. (2024)
over mainland China, and revealed the significant improvement of the gauge corrected versions compared to the near-real-
time products. Furthermore, Haile et al. (2013) assessed CMORPH and TRMM datasets against gauge observations for the
diurnal cycle of precipitation over the Nile River basin, noting over-detection over lake surfaces and underestimation over
mountaintops. Similarly, Zhang et al. (2021) revealed the comparatively better performance of CMORPH followed by TRMM
and PERSIANN over the Three Gorges Reservoir area in China at 6h and 12h scales. All three estimates are in close agreement
with the observations in terms of the diurnal cycle, though PERSIANN exhibits some biases. Moreover, Roca et al. (2021)
provided an overview of the uncertainties at the hourly scale among the satellite estimates at their native resolution across
specific regions of the world.

In addition to satellite products, reanalysis datasets, such as the ECMWF Reanalysis v5 (ERA5) (Hersbach et al., 2020), also
have the capability of representing the diurnal cycle of precipitation at the global level. Indeed, the application of reanalysis
datasets in studying the diurnal cycle of precipitation has extensively increased over the years (Chen et al., 2014; Jiang et al.,
2021; Qin et al., 2021). Several attempts have been made to evaluate the performance of ERA5 at various spatial-temporal
scales (Nogueira, 2020; Beck et al., 2019). For instance, Beck et al. (2019) report that ERA5 has shown better performance
than IMERG in precipitation estimation in complex regions (mountainous terrains); however, the opposite is true in regions
characterised with short-lived convective systems. In contrary, Sharifi et al. (2019) reported that IMERG outperforms the ERA5
at the complex terrain on the daily and monthly scale over Austria. Studies have also attempted to evaluate the ERA5 at a sub-
daily scale (Kumar et al., 2021; Hong et al., 2021; Tang et al., 2020) and most of them find ERA5 has difficulties in estimation
of diurnal cycle of precipitation when compared to IMERG.

Despite the increasing number of sub-daily scale precipitation studies at the regional scale, such studies are rarely available
at the global level. Moreover, to our knowledge, there have not been many studies that evaluated the various satellite estimates
in terms of their diurnal variation and their performance at sub-daily scales globally (Dai et al., 2007). Here, we address this
gap. We compare five state-of-the-art precipitation estimates, IMERG, GSMaP, PERSIANN, CMORPH, and ERA5, at their
hourly scales. This study represents the first comprehensive analysis of the global diurnal cycle, leveraging an ensemble of
current satellite estimates compiled from two decades of datasets. This will enable us to examine the region-specific strength
and limitation of the precipitation estimates.

The paper is organized as follows. Section 2 introduces the five datasets used in the analysis as well as the methodological
approaches employed. Then, in Section 3, we present the findings of the analysis, starting with the spatial mean precipitation
across the globe and their zonal distribution, and following with the diurnal cycle and its variation across the globe and among
the datasets. In Section 4, we focus on the underlying mechanisms responsible for the observed results, along with some limi-
tations and future perspectives. Finally, we conclude this study, highlighting what we have learned from this intercomparison,
in Section 5.





## 2 Data and Methodology

**Satellite-based datasets**

The Global Precipitation Measurement (GPM) mission is a constellation of international satellites that aims to provide high-quality precipitation observations with quasi-global coverage (Huffman et al., 2015). IMERG is a unique algorithm that merges and inter-calibrates precipitation estimates from a range of sources, such as Passive Microwave (PMW), Infrared (IR), and gauges in order to produce $0.1° \times 0.1°$ and 30 min precipitation products (Huffman et al., 2020). Unlike the near-real-time products, IMERG Final run (IMERG-F) incorporates ground gauge correction from Global Precipitation Climatology Centre (GPCC) estimates. IMERG-F products are mainly intended for research purposes and are available after 3.5 months of observation. A substantial number of studies have validated the IMERG performance in a range of climatic conditions, and it performed extremely well in a wide range of applications (Pradhan et al., 2022). Further, more details regarding the IMERG precipitation estimation algorithms and their technical details can be found in Huffman et al. (2015). IMERG V06B Final Run Half Hourly product is used in this study.

GSMaP is a gridded multi-satellite precipitation product developed jointly by the Japan Aerospace Exploration Agency (JAXA) and Japan Science and Technology Agency (JSTA) (Kubota et al., 2020; Mega et al., 2019). GSMaP merges precipitation estimates from a range of several low earth orbit passive microwaves and geostationary IR precipitation sensors. Like IMERG, GSMaP also provides precipitation in near-real-time, as well as gauge-corrected final products. Nonetheless, unlike IMERG, GSMaP uses the Climate Prediction Centre (CPC) unified global daily gauge precipitation for gauge correction. Moreover, validation studies have reported consistent performance of GSMaP with observations (Zhou et al., 2020; Lu and Yong, 2018), and relatively better than IMERG at least in a few cases (Li et al., 2021; Ning et al., 2017; Salles et al., 2019). In the present study, we have used the GSMaP gauge corrected V08 product, available at hourly and $0.1° \times 0.1°$ resolutions.

CMORPH products utilize passive microwave estimates from low earth orbiting satellites to generate high-quality global precipitation estimates (Joyce et al., 2004; Joyce and Xie, 2011). Given the low-sampling nature of microwave estimates, CMORPH incorporates geostationary Infrared (IR) image-derived information to propagate precipitation systems (i.e., CPC Morphing technique) in instances where microwave data is unavailable. Additionally, CMORPH integrates CPC daily precipitation estimates over land and GPCP pentads merged analysis over the ocean for bias correction (Xie et al., 2017). Although CMORPH offers very high spatial resolution, specifically 0.07277 degrees latitude/longitude, this study utilizes the CMORPH bias-corrected Version 1 (V.1) product with a spatial resolution of $0.25° \times 0.25°$ and hourly temporal resolution, as accessed from https://www.ncei.noaa.gov/data/cmorph-high-resolution-global-precipitation-estimates/access/hourly/

PERSIANN, primarily relies on the geostationary infrared cloud images to estimate precipitation at $0.25° \times 0.25°$ spatial resolution and hourly temporal scales (Hsu et al., 1997; Sorooshian et al., 2000). As its name suggests PERSIANN uses artificial neural networks to estimate precipitation based on the cloud top temperature from the geostationary satellite-derived infrared images (Nguyen et al., 2019). It is important to note that PERSIANN does not directly incorporate passive microwave (PMW) estimates as input; instead it, uses LEO satellites to continually adjust the parameters of the model. The PERSIANN products used in the current study are obtained from the CHRS Data Portal at https://chrsdata.eng.uci.edu.





**Reanalysis estimates-ERA5**

ERA5 is the latest fifth-generation global atmospheric reanalysis product developed by the European Center for Medium-Range Weather Forecasts (ECMWF) using the 4D-Var data assimilation techniques in cycle 41r2 (Hersbach et al., 2020; Jiang et al., 2021). Recently, ERA5 replaced its predecessor, i.e., the ERA-Interim reanalysis product. Compared to ERA-Interim, ERA5 has been updated with a more advanced data assimilation system and physical model, and more importantly, the spatial resolution is reduced to 31 km. In addition, ERA5 datasets are now available at an hourly scale and have extended to 1950. Compared to other global reanalysis products, ERA5 has better agreement with the observational precipitation (Keller and Wahl, 2021; McClean et al., 2023), and reported substantial improvement compared to the ERA-interim (Wang et al., 2018). Here, we have used the hourly ERA5 reanalysis data from 2001 - 2020.

**Table 1.** Summary of the datasets used in this analysis.

| Dataset name | Spatial scale | Temporal scale | Record length | Reference |
|---|---|---|---|---|
| IMERG | $0.1° \times 0.1°$ | 0.5h | 2000 – present | Huffman et al. (2015) |
| GSMaP | $0.1° \times 0.1°$ | hourly | 2000 – present | Mega et al. (2019) |
| ERA5 | $0.25° \times 0.25°$ | hourly | 1950 – present | Hersbach et al. (2020) |
| PERSIANN | $0.25° \times 0.25°$ | hourly | 2000 – present | Hsu et al. (1997) |
| CMORPH | $0.25° \times 0.25°$ | hourly | 1998 – present | Joyce et al. (2004) |

**Methodology**

The methodological approach includes an inter-comparison of sub-daily scale precipitation derived from multiple sources of precipitation datasets (Table 1). The multi-source precipitation datasets from the state-of-the-art satellite and reanalysis products at their original resolution (i.e., 30 minutes or hourly) are collected covering the global land and ocean between $60°N – -60°S$. Considering the different temporal coverage of each dataset, a common overlapping period from 2001 – 2020 is selected as the analysis period. Moreover, given the different spatial and temporal resolutions of the datasets, to have a consistent and fair analysis, all the estimates are converted into a common spatial and temporal resolution of $0.25° \times 0.25°$ and hourly scale. For datasets such as IMERG and GSMaP, which have resolutions finer than $0.25°$ (e.g., $0.1° \times 0.1°$ native resolution), a multi-step conversion process is employed. Initially, these datasets are upscaled to the closest target resolution (i.e., $0.2° \times 0.2°$) by means of simple averaging. Subsequently, they are further refined to the desired $0.25° \times 0.25°$ resolution through the nearest neighbour interpolation technique.

The sub-daily scale evaluation among the estimates is based on the diurnal cycle. According to Watters et al. (2021), the first and second-order harmonics are often not efficient in capturing the diurnal variability. Therefore, in this study, we are not fitting any harmonic function or empirical orthogonal function to estimate diurnal parameters. Instead, the diurnal variability of global precipitation is evaluated focusing on three parameters key to our understanding of precipitation, namely, the precipitation





amount, the frequency of wet time intervals, and the average intensity of the wet time intervals (Marzuki et al., 2021). The
mean precipitation amount is estimated by accumulating all the hourly precipitation divided by the total available hours for
each grid. For frequency, the total number of precipitating hours (precipitation > 0.1 mm/hr) is divided by the total available
hours (both dry and wet). Finally, the intensity is estimated with the total precipitation divided by the precipitating hours
(precipitation > 0.1 mm/hr). As mentioned above, all these metrics are estimated for each grid.

In this study, we defined a wet hour, or rainy hour, as one with precipitation equal to or greater than 0.1 mm/hr. This threshold
is commonly employed in similar analyses and has been widely utilized in previous research (Dai et al., 2007; Xiao et al.,
2018). It is well known that, the frequency and intensity of precipitation events are highly sensitive to the selected thresholds.
Therefore, to explore the impact of different threshold levels on frequency and intensity, we conducted additional analyses
using alternative thresholds, specifically 0.2 mm/hr and 0.5 mm/hr (Figure. S4, S5, and S6). While the overarching findings of
our study were not influenced (the diurnal pattern remains similar) by the specific threshold chosen, this investigation provided
valuable insights into how varying threshold criteria can affect the results.

The mean precipitation amount for each latitude ($\phi$) and longitude ($\lambda$) at the Universal Coordinated Time (UTC) time hour
($t_{UTC}$) is estimated by using the following equation (Eq. 1)

$$P(\phi, \lambda, t_{UTC}) = \frac{\sum_{i=1}^{N} Pi}{N} \tag{1}$$

Here, $P_i$ represents the $i$th precipitation estimate of the study period, and N represents the total number of precipitation
estimates for a given latitude, longitude and hour (including no precipitation events).

The UTC hour of each precipitation dataset is converted to Local Solar Time (LST) by using the following equation (Eq. 2)

$$t_{LST}(h) = t_{UTC}(h) + \frac{\lambda(°)}{15(°h^{-1})} \tag{2}$$

Given that the analysis is conducted at the level of each grid, we acquire unique diurnal patterns for each grid. However,
visualizing these patterns on a global scale becomes unfeasible due to the sheer volume of data. To address this challenge, we
employ the K-means clustering algorithm. This method enables us to group similar diurnal patterns together, providing a more
condensed representation of global diurnal precipitation variability. In this study, we utilized the hourly mean precipitation
amount as the variable for clustering to identify pixels with similar diurnal cycles. Prior to performing the K-means clustering,
the mean precipitation data undergoes normalization to ensures that the K-means clustering process is not influenced by vari-
ations in precipitation levels across different regions. This normalization process involves scaling the precipitation values by
their mean and standard deviation at each grid point. To determine the optimal number of clusters, the process is iterated from
K = 1 to K = 15, ultimately selecting k = 4 as the appropriate number of clusters because the reduction in terms of within sum
of squares (WSS) including additional clusters was below 7%. Subsequently, four clusters are extracted from each dataset to
depict global diurnal precipitation variability. Finally, each cluster is named according to its peak hour of local solar time.



# 3 Results

## 3.1 Spatial distribution of mean hourly precipitation properties

In order to investigate the differences and similarities among precipitation products, first, we examined the spatial distribution of mean hourly precipitation amount, frequency, and intensity at $0.25° \times 0.25°$ resolutions for the period of 2001 – 2020. The distribution of hourly mean precipitation exhibits a consistent spatial pattern among the estimates across the globe (Figure 1, and differences from the ensemble can be seen in Figure S1). In particular, visually, all of them depict similar spatial patterns characterized by high precipitation across the Intertropical Convergence Zone (ITCZ) and South Pacific Convergence Zone (SPCZ) belt, and low precipitation over the dry regions in the subtropical high and across the Sahara desert. As mean hourly precipitation is directly related to the total precipitation amounts, it is not surprising to have similar pattern across the datasets. However, regional differences in the dry regions (e.g., southern Pacific Ocean, Southern Atlantic Ocean, and southern Indian Ocean near Australia) can be observed between the PERSIANN and ERA5. Unlike other data products, PERSIANN exhibits more widespread dry regions, while the opposite is true for ERA5. Given the known limitations of infrared (IR)-based estimates in accurately detecting precipitation from warm clouds (Behrangi et al., 2012), the underestimation of precipitation by PERSIANN over tropical oceans, where warm rainfall is prevalent, could be attributed to this factor. Likewise, the presence of a wet bias in climate model simulations has been observed across different regions (Ou et al., 2023), which may account for the widespread occurrence of wet conditions in ERA5.

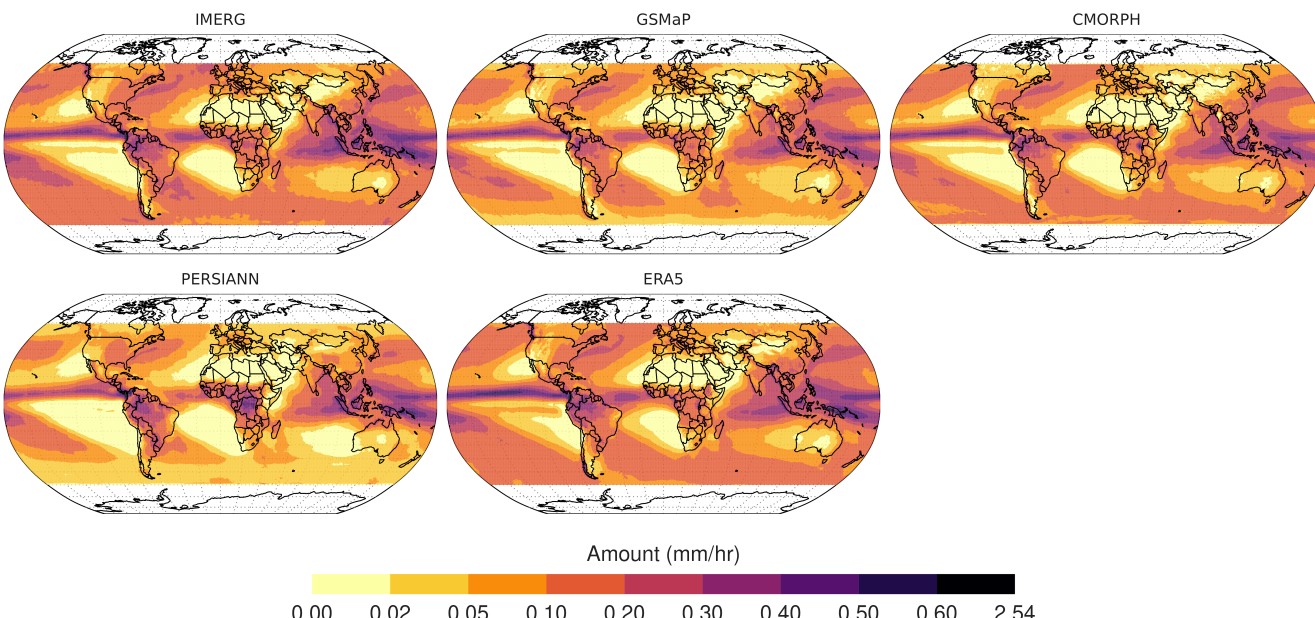

**Figure 1.** Spatial distribution of global mean (2001 – 2020) hourly precipitation amount (mm/hr).





In terms of the spatial distribution of hourly precipitation frequency, there is a generally similar pattern among the estimates (Figure 2), and it resembles those of mean precipitation amounts (Figure 1). Nonetheless, ERA5 appears markedly different from the remote sensing data products, showing substantially high frequencies across the globe. In particular, across the tropical belts and, more precisely, over the ocean, ERA5 shows substantially higher precipitation frequency (40 – 90%) than the rest of the estimates. Similar to the mean precipitation amount, the frequency of hourly precipitation appears to be quite low
in PERSIANN, particularly over dry regions such as the subtropical high and the Sahara regions of the African continent. Additionally, compared to the IMERG and GSMaP, CMORPH also exhibits relatively lower frequencies across the globe.

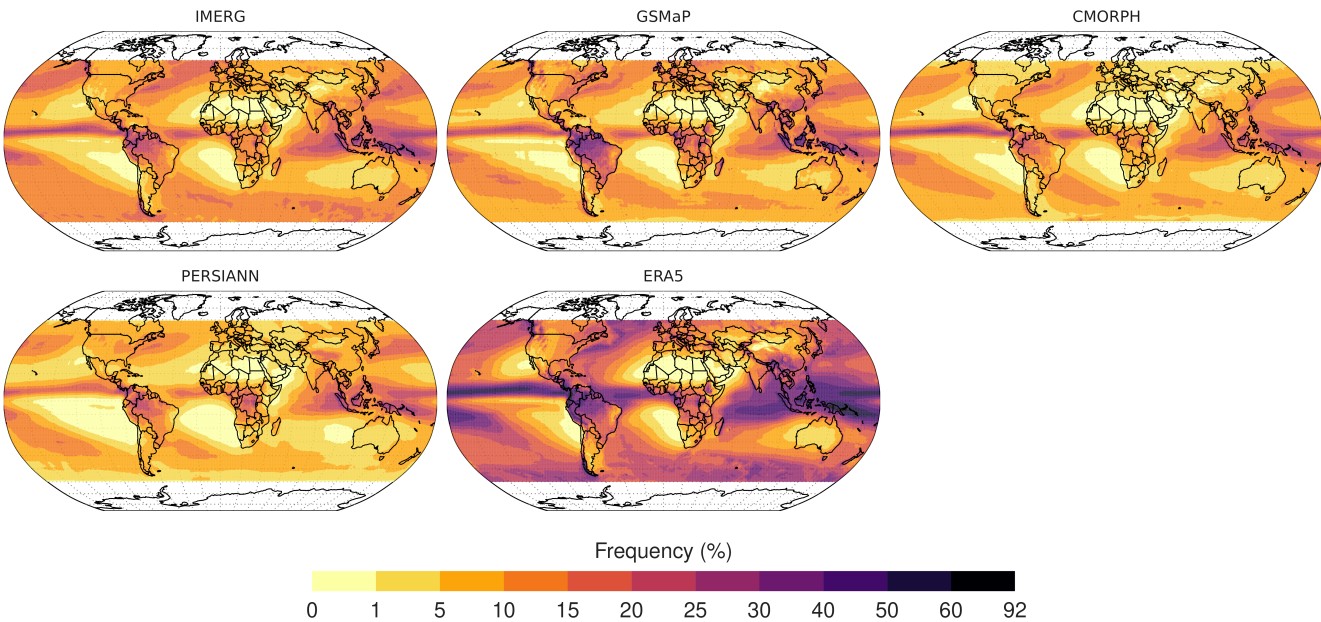

**Figure 2.** Same as Figure 1, but for precipitation frequency (%).

Unlike the mean precipitation amount and frequency, the spatial pattern of mean precipitation intensity is not very homogeneous among the datasets (Figure 3). The mean intensity of precipitation seems comparatively very low in the ERA5 estimates compared to the rest of the datasets. This confirms that model reanalyses tend exhibit a high-frequency, low-intensity issue,
a concern that has been extensively reported over the years (e.g., Watters et al. 2021; Qin et al. 2021). In terms of satellite estimates, there is a relatively good agreement between the IMERG and CMORPH throughout the globe, compared to the GSMaP and PERSIANN. In fact, surprisingly the mean intensity of precipitation for GSMaP also appears to be relatively low, especially over global land. Moreover, it can also be noted that CMORPH exhibits relatively higher intensities, which could be attributed to the relatively lower fraction of light precipitation events in the CMORPH.
To further explore and compare the various estimates, the latitudinal average of the mean precipitation amount, frequency, and intensities are also examined (Figure 4a, 4b and, 4c). In terms of hourly mean latitudinal precipitation, all the products





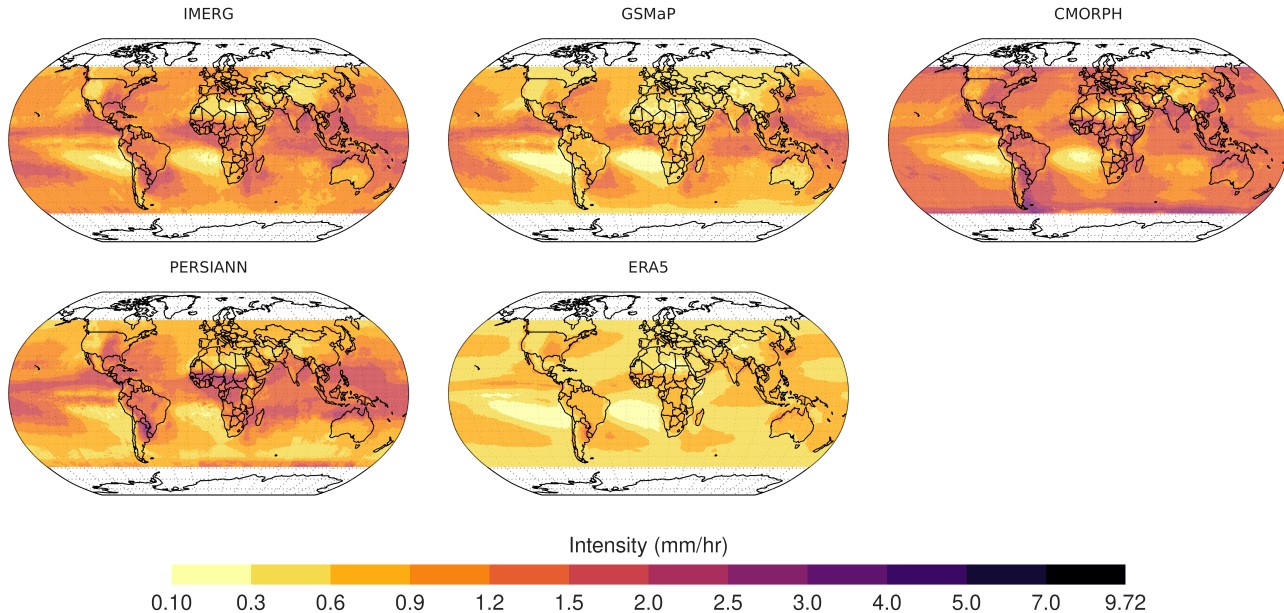

**Figure 3.** Same as Figure 2, but for precipitation intensity (mm/hr).

agree and exhibit a similar pattern with a peak at the ITCZ followed by a minimum in sub-tropical belts and so on (Figure 4a). At the ITCZ, and between 0°N to 10°N, in particular, all datasets accurately depict the peak. However, ERA5 (0.3 mm/hr) and IMERG (0.27 mm/hr) show close estimates, which are distinguished from the ones of the other products (values around 0.25 mm/hr). Moreover, all the datasets have similar values from 35°N to -20°S. At higher latitudes, the differences among the estimates increase with latitude. Furthermore, compared to the northern hemisphere, the variability among the datasets is notably higher in the southern hemisphere (-20°S – -60°S). This could be related to the lower availability of ground observations that are used to calibrate and adjust satellite products and are assimilated in the reanalysis simulations. Additionally, while ERA5 exhibits the highest mean precipitation amounts with a peak at the ITCZ, IMERG surpasses ERA5 in extra-tropical belts (-20°S to 20°N) in the southern hemisphere. Conversely, in the northern hemisphere, ERA5 retains the highest mean precipitation amounts. GSMaP aligns closely with CMORPH and PERSIANN, in contrast to IMERG. In the southern hemisphere, GSMaP has shown a sharp decline from -30°S until -60°S.

In terms of frequency, ERA5 exhibits a spatial pattern similar to other datasets throughout the latitudes (Figure 4b). However, its frequency estimation is often significantly higher across all latitude zones. At the ITCZ belt, where the peak frequency occurs, ERA5 has a frequency reaching up to 40%, which is almost double that of the rest of the products (< 20%). Furthermore, among the latitudinal zones, 20° – 30° is the only region in both the hemispheres, where the difference between ERA5 and the rest of the datasets is relatively minimal. When it comes to the remote sensing estimates, IMERG and GSMaP have a very close agreement throughout the latitudinal zones. However, from -40°S onward, the difference between IMERG





and GSMaP keeps increasing with the latitude. GSMaP exhibits a sharp decline in a manner similar to the mean precipita-
tion amount (Figure 4a). Although the PERSIANN and CMORPH are in close agreement throughout the latitudinal zones,
PERSIANN remained lowest among the estimates, particularly in the southern hemisphere.

As a consequence of the high frequency, the intensity of ERA5 remains the minimum among the datasets throughout the
latitudinal zones (Figure 4c). In fact, the highest uncertainties among the estimates are observed in terms of representing the
precipitation intensity. Again, compared to the northern hemisphere, the discrepancies are highest over the southern hemi-
sphere, with the highest occurring towards the higher latitudes. PERSIANN shows the highest intensity over the ITCZ belts
(-20°S to 20°N), with values up to 1.5 mm/hr, followed by CMORPH, IMERG, and GSMaP. However, from the extra-tropical
regions, especially from 20°N/S on-wards, CMORPH shows the highest intensity with an increasing trend with the latitudes,
which is quite the opposite of the rest of the products. Despite having low frequencies, both PERSIANN and CMORPH show
high intensity, probably due to the missing light precipitations.

Overall, all the datasets effectively capture the spatial variability and distribution of mean precipitation, frequency, and
intensity. However, the zonal plots reveal significant differences among them, especially towards the high latitudes. In fact,
the challenge of precipitation retrievals toward high latitudes has been reported in previous studies as well (Protat et al., 2019;
Grecu et al., 2016; Skofronick-Jackson et al., 2017). ERA5 exhibits the highest frequency and lowest intensity, while CMORPH
depicts the highest intensity. The remaining datasets fall in between, contributing to the observed variations.

## 3.2 Diurnal variation

### 3.2.1 Diurnal mean

The diurnal variation of mean precipitation amount among the datasets is quite similar in shape (Figure 5a). In other words,
all the products agreed well in terms of producing specific features of the diurnal variation over the globe: an afternoon peak
over the land and an early morning peak over the ocean. In addition, a bimodal peak with peaks in the early morning (from the
ocean) and afternoon (from land) can be observed at the global level. These diurnal results are consistent with previous studies
(Dai et al., 2007; Watters et al., 2021). However, significant differences exist among the estimates as well. At the global level,
IMERG and ERA5 look quite close to each other with the highest values, whereas GSMaP and PERSIANN have the lowest
and CMORPH is in between. Over the ocean, the behaviour of the products is also quite similar to the global level, except for
one difference: the early morning peak. This also indicates that the ocean diurnal cycle dominates at the global level, which is
expected as the ocean receives the lion's share of global precipitation compared to land.

Over the land, all the products well reproduce the afternoon peak, a common feature of the diurnal cycle. This is consistent
with other studies over the years. ERA5 shows the peak a little earlier, around 15 LST over land, compared to the other estimates
which are mostly between 16 to 18 LST. The earlier peak from the ERA5 reanalysis and other model-generated precipitation
is, in fact, not uncommon (Hayden et al., 2023). Additionally, ERA5 also shows the highest peak with a mean precipitation
of around 0.18 mm/hr, followed by PERSIANN (0.15 mm/hr), IMERG (0.14 mm/hr), CMORPH (0.11 mm/hr) and GSMaP
(<0.1 mm/hr). While ERA5 shows the peak and diurnal cycle slightly earlier than other datasets, the uncertainty among them





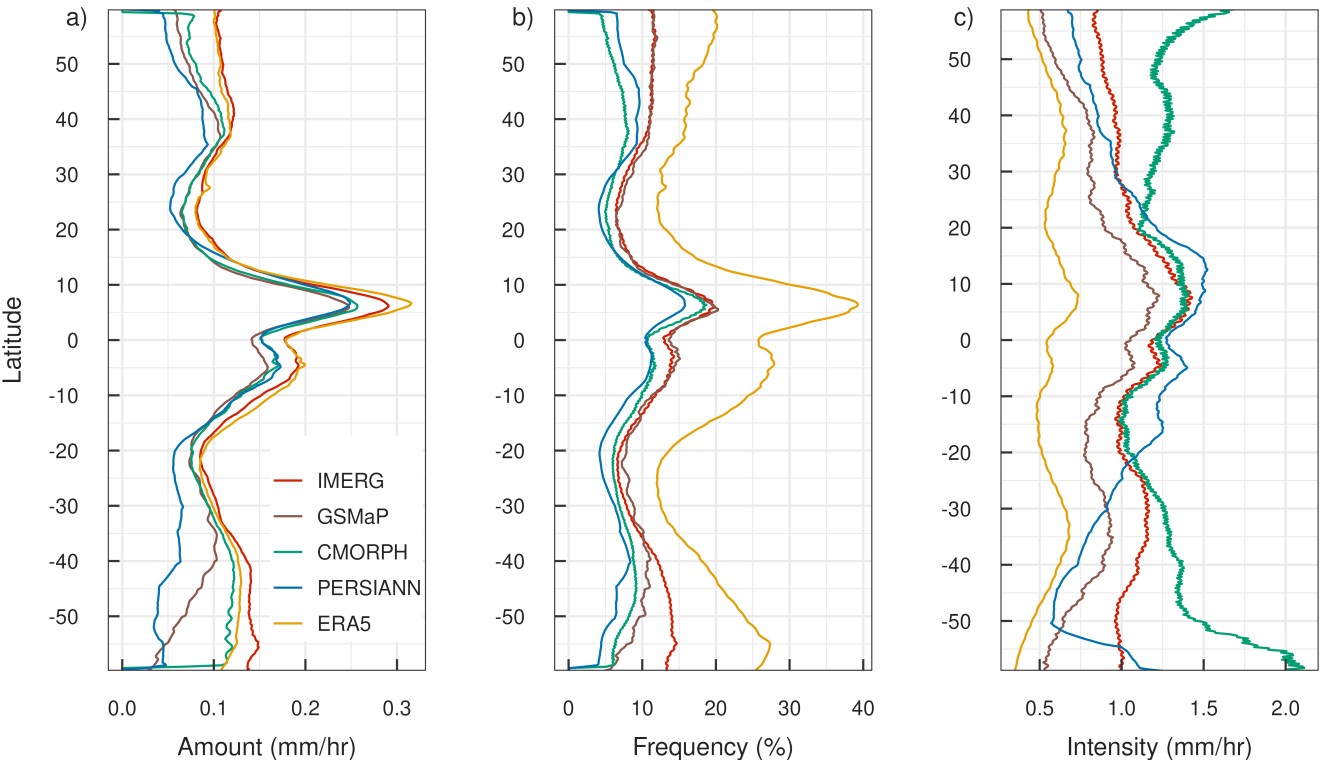

**Figure 4.** Latitudinal average of mean hourly precipitation a) amount (mm/hr), b) frequency (%), and c) intensity (mm/hr).

is greater between 11 to 18 LST, whereas it is minimal during other times, such as early morning and night. This highlights the different performance of retrieval algorithms, temporal sampling and model simulations at observing or reproducing convective precipitation.

### 3.2.2 Diurnal frequency

The diurnal variation in precipitation frequency across the datasets looks quite different compared to the mean precipitation amount (Figure 5b). As expected, ERA5 displays considerable deviation from the other datasets, irrespective of the region (i.e., globe, land, ocean). Despite the significant overestimation, ERA5 exhibits a peak during the afternoon around 16 LST, aligning with the peak observed in other satellite precipitation datasets over land. At the global level, and over the ocean, the variation in diurnal frequency is not particularly pronounced across the satellite datasets. However, ERA5 shows a distinct peak in frequency early in the morning (between 03 to 04 LST), with the diurnal variation being more predominant over the ocean. All the other estimates show little variation (a flatter shape) with multiple peaks throughout the 24-hour period. On the contrary, over land, the peak of diurnal frequency is between 15 LST to 16 LST. Regarding amplitude, while all datasets show similar peaks in the afternoon, substantial disparities among them are apparent. ERA5 exhibits relatively high discrepancies





**Figure 5.** Diurnal variation of precipitation a) amount (mm/hr), b) frequency (%), and c) intensity (mm/hr), of each dataset for 2001 – 2020.

(exceeding 20%) during the daytime from late morning (08 LST) until late evening (19 LST). In terms of the satellite products, their agreement varies with the region. At a global level, IMERG and GSMaP appear similar, as do CMORPH and PERSIANN. Over land, notably, GSMaP is comparatively high, followed by IMERG, PERSIANN, and CMORPH.

### 3.2.3   Diurnal intensity

Unlike the diurnal mean precipitation amount and frequency, the diurnal intensity is not so pronounced, and thus, it does not

exhibit a clear diurnal variation or pattern (Figure 5c). Peaks are not distinctly evident globally and over the ocean. However, over land, a bimodal peak can be seen, with an early morning peak between 00 – 05 LST, and a late afternoon peak at 15 – 21 LST.

In terms of different precipitation products, it is evident that CMORPH exhibits the highest intensity (exceeding 1.25 mm/hr) regardless of whether it is over land, ocean, and globally throughout the 24-hour period. IMERG ranks second in terms of the





highest intensity over global and oceanic levels, whereas it is PERSIANN over land. As expected, ERA5 exhibits the minimum

intensity throughout the day, with a very weak diurnal variation over the ocean and globe compared to land. Despite having a

similar pattern, GSMaP does not follow IMERG and has the lowest precipitation intensity over land, even lower than ERA5.

GSMaP consistently has the lowest intensity after ERA5, whether over global, land or ocean regions. It also has the largest

discrepancy with the other datasets, particularly over land. This behavior of GSMaP over land is notable, considering both

IMERG and GSMaP use a similar constellation of satellite estimates. Nevertheless, it should also be noted that both datasets

use different gauge corrections over land. IMERG uses Global Precipitation Climatology Centre (GPCC) corrections on a

monthly scale, while GSMaP uses CPC corrections on a daily scale, and this could be the likely reason for such observational

differences between the datasets.

### 3.3    Peak hour of diurnal mean precipitation amount, frequency, and intensity

To further investigate the timing of maximum precipitation properties (i.e., amount, frequency, and intensity) and their varia-

tions across different climates and topographic regions, the peak hours are also examined (Figure 6, 7, and 8). The peak hour

denotes the hour at which the maximum precipitation properties occur at each grid. Regarding the peak hour of precipitation

amount, the continental/land regions are mostly dominated by the evening peak hours (15 – 18 LST), compared to the early

morning peak hours over the ocean (02 – 06 LST) (Figure 6). However, there are regions, particularly over land, where a

slightly inhomogeneous distribution of peak precipitation hours is observed. This inhomogeneous distribution is mainly seen

in dry regions such as Africa, Australia, and the Middle East. Topographic barriers appear to have an impact on the timing of

peak precipitation hours. In regions upwind of high-elevation mountain chains such as the Himalayas and Andes, peak hours

tend to occur in the early morning, in contrast to the early afternoon peak hours observed in the surrounding land (Figure

S7). This phenomenon is likely due to orographic precipitation. In China, similar results were observed; convective systems in

eastern Asia typically develop in the foothills of the Tibetan Plateau during midnight and propagate eastwards during late night

to early morning (Chen et al., 2017).

The oceanic regions are mainly characterized by midnight to early morning peak hours (00 – 06 LST). In fact, the high

precipitation regions are mainly dominated by early morning peak hours, approximately between 03 – 07 LST, while the dry

regions, such as the Atlantic and Pacific warm pools, are a bit earlier, between 01 to 03 LST. It is even earlier, around 22 to 01

LST, towards the high latitudes (pole wards). In particular, this pattern is more pronounced over the southern hemisphere, as

clearly observed in IMERG and CMORPH datasets, although not as distinctly captured by GSMaP.

In the coastal regions near the land, precipitation peaks in-between, i.e., 06 – 12 LST, and as it progresses towards the land,

the peak hours keep increasing and reach the typical late-afternoon/early evening peaks (15 – 18 LST). The exact opposite

pattern is observed towards the ocean with precipitation peak in the early morning (03 – 06 LST), although regional differences

exist among the estimates (e.g., over the Southern Ocean). Similar results were also observed in previous studies, such as by

Hayden et al. (2023) and Bai and Schumacher (2022) over the maritime continents.

In terms of agreement among different datasets, most of them portray similar spatial patterns, although some noticeable

differences exist among them. In particular, ERA5 differs slightly from the rest of the datasets, with an early peak hour over





both the land and oceanic regions. PERSIANN is quite different from the other estimates, which is expected as it is based
on different sensors and retrieval principle. IMERG and CMORPH show a high degree of similarity, while GSMaP depicts
noticeable differences. In particular, IMERG and CMORPH agree well, not only in producing peaks in the early morning
hours over the ocean and in the late afternoon over land, but they are also consistent in depicting the small regional differences.
For instance, both IMERG and CMOPRH agree on the occurrence of midnight peak hours in the high-latitudes Southern Ocean
and various dry regions, such as the sub-tropical region of the southern Atlantic and Pacific, as well as over Africa and the
Middle East. This agreement could be due to the fact that both IMERG and CMORPH have relied on microwave estimates,
and, more importantly, the CMORPH algorithm is partially incorporated in the IMERG.

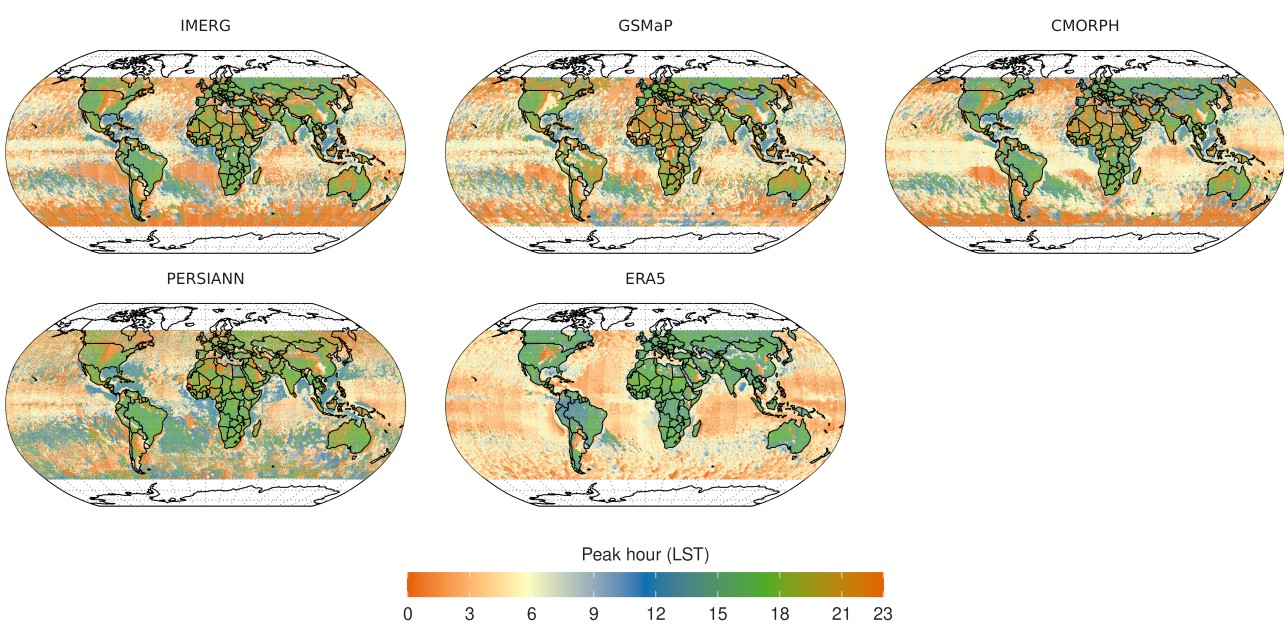

**Figure 6.** Peak hour of mean precipitation for each dataset during 2001 – 2020.

Over land, similar to the peak hour of mean precipitation amount, precipitation frequency peaks during the afternoon between
14 – 18 LST (Figure 7). However, over the ocean, depending on the region, it varies with time and datasets. The discrepancies
among the estimates are larger over the ocean than over land. For instance, as observed by IMERG, sub-tropical dry regions
such as the Pacific and Atlantic warm pool zones show nighttime frequency peaks around 02 – 03 LST, while in the high
precipitation ITCZ belts, peaks occur during the day (11 – 12 LST). In the high latitude zones (40° – 60°), the northern
hemisphere experiences midnight peaks (22 – 01 LST), whereas the southern hemisphere peaks during midday (10 – 12
LST). All these differences in peak hour over the ocean by IMERG are not consistent with other datasets. However, IMERG
and CMORPH agree over the high latitude northern oceans (> 40°N) with nocturnal peaks during 21 – 02 LST. Nonetheless,
discrepancies arise on the southern oceans, as IMERG shows late-morning/midday (09 – 12 LST), while CMORPH is nocturnal





with few midday peaks. In fact, GSMaP also depicts a similar pattern; however, it is observed in the early morning (03 – 06 LST), and the nocturnal peaks are restricted to the high latitudes ($> 50°$N).

In terms of different datasets, similar to the peak hour of mean precipitation amount, all the remote sensing estimates are consistent in reproducing the nocturnal peaks (21 – 01 LST) over the Great Plains in the United States, southern Brazil, central
and northern Africa, eastern China, and parts of Australia. These region-specific features, however, are not produced by ERA5, and overall, it does not show the spatial variability in peak hour frequency. Instead, ERA5 exhibits a more uniform pattern with peak frequency hours predominantly varying between 00 to 06 LST, showing little distinction whether they occur in the polar oceans or in the tropical regions.

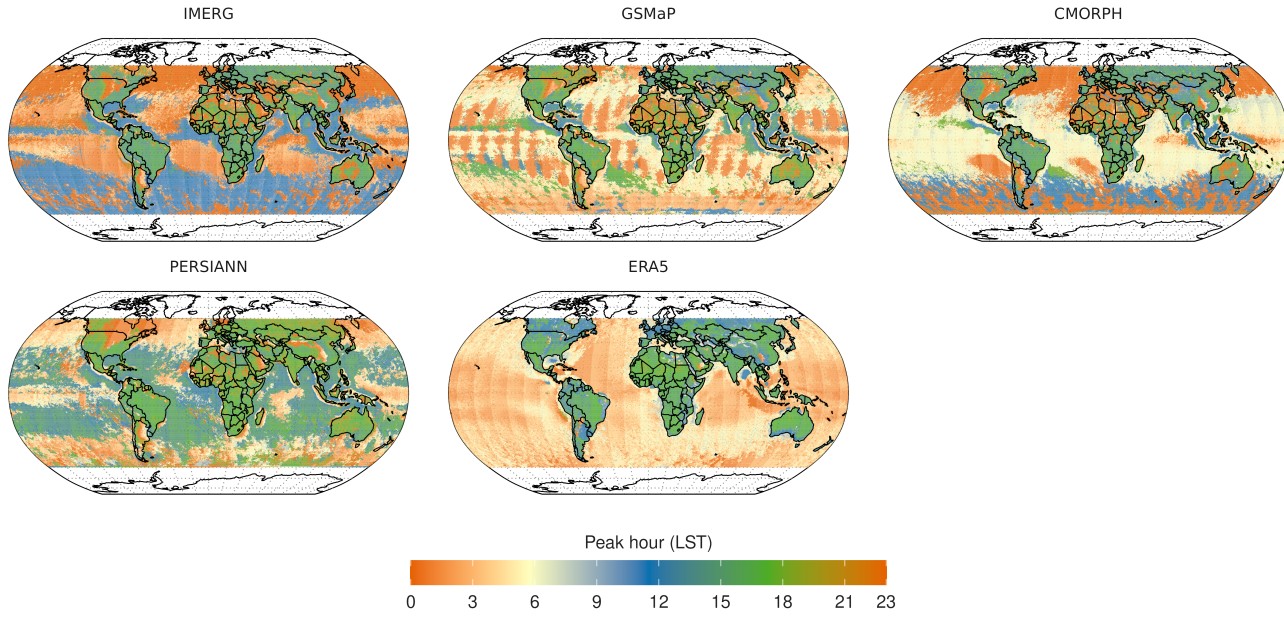

**Figure 7.** Peak hour of precipitation frequency for each dataset during 2001 – 2020.

Unlike the peak hour of mean precipitation amount and frequency, the peak hour of intensity exhibits significant hetero-
geneity over both land and ocean (Figure 8). The difference in peak hours of intensity between land and ocean is not very pronounced. Over land, the peak hour of intensity occurs either during the late night/early morning (02 – 06 LST) or late afternoon between 15 – 18 LST, depending on the region and the dataset. This observation highlights that although the mean and frequency of peak hours predominantly occur during the late afternoon over land, high-intensity precipitation mainly occurs during late nights or early mornings, though some regions also exhibit late afternoon peaks as well. Despite all the satellite
estimates being consistent in depicting the regional pattern, slight differences exist among them. When compared to IMERG, CMORPH exhibits a similar pattern, whereas GSMaP exhibits a slight delay. In PERSIANN and ERA5, the peak hours are further delayed, and hence the majority of the land regions depict the peak hours during the late afternoon between 15 – 18



LST. Especially in regions such as northern South America (Brazil), southern Africa, and Canada and Russia in the northern hemisphere, ERA5 depicts afternoon peaks. In contrast, satellite estimates depict a mix of peak hours ranging from late
night/early morning to a few afternoon peak hours. Unlike the peak hour of mean and frequency, ERA5 is consistent with satellite estimates, producing regional differences in peak hour intensity in regions such as the Great Plains of the USA, the southern region of South America (Peru), central regions of Africa, and parts of Australia.

    Over the ocean, all satellite estimates indicate an early to late morning peak between 03 – 09 LST. In ERA5, however, the peak extends from late morning to early afternoon, covering the period from 09 – 15 LST. IMERG exhibits an early morning
peak between 03 – 06 LST throughout most of the tropical oceans and the majority of the northern hemisphere. Towards the poles and in most of the southern hemisphere, it shows late evening peaks between 06 – 08 LST. Despite a similar pattern between IMERG and GSMaP, GSMaP exhibits the peak hour slightly later, occurring between 07 – 09 LST and 19 – 21 LST. CMORPH, on the other hand, shows peak hours falling between the range of IMERG and CMORPH. Regarding PERSIANN, it exhibits peak hours even earlier than the other satellite estimates, with the majority of the ocean showing peaks between
03 – 05 LST. Among the datasets, ERA5 shows the greatest deviation, with most oceanic regions exhibiting peaks in the late morning to early afternoon (09 – 15 LST). The only exception is in the Indian Ocean and western Pacific regions, where ERA5 shows early morning peaks between 06 – 09 LST, which is consistent with the other datasets. These results are not surprising, given the significant differences among the datasets, as illustrated in Figure 5c.

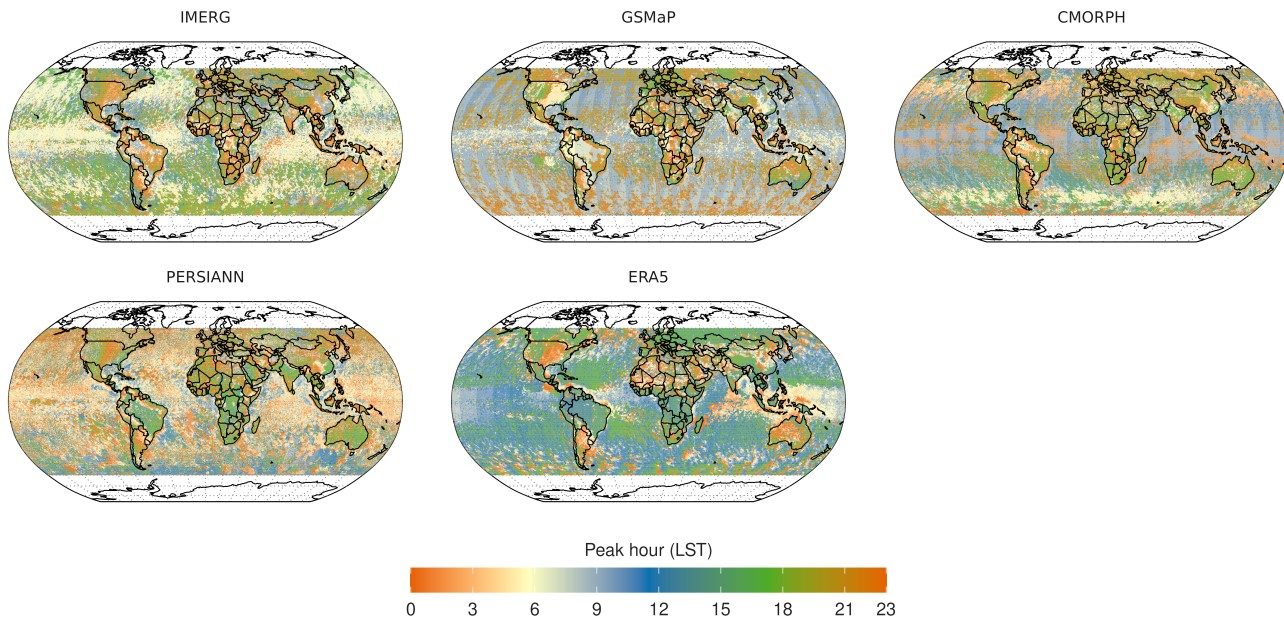

**Figure 8.** Peak hour of precipitation intensity for each dataset during 2001 – 2020.



### 3.4 Spatial distribution of diurnal characteristics

Although it is observed how diurnal precipitation varies between datasets in terms of land, ocean and globe, it does not provide information on how these shapes change at each grid level. To accomplish this, the K-means clustering algorithm is employed. (Figure 9). The clusters are named according to their respective peak hour of local solar time: afternoon peak (red), early morning peak (grey), late morning peak (green), midnight peak (yellow), and early afternoon peak (blue).

Unsurprisingly, both IMERG and CMORPH produce similar clusters, while GSMaP and PERSIANN fall in between. ERA5
seems to represent a distinct climatology. All datasets share an afternoon peak hour cluster, with a maximum between 15 – 17 LST and a minimum between 09 – 10 LST, although there are noticeable differences in magnitude. In contrast, the early morning peak, at 08 – 10 LST, is another cluster that can be seen in all the products. The early afternoon peak, observed over the GSMaP, PERSIANN, and ERA5 with peaks between 14 – 15 LST, has quite different amplitudes among the estimates. ERA5 shows the highest amplitude ($> 0.20$ mm/hr), whereas it remains $< 0.10$ mm/hr in GSMaP and PERSIANN. The peak in the
late morning, which closely resembles the early morning peaks, is delayed by one hour and is present in all estimates except for PERSIANN. Conversely, the peak at midnight has been observed in all estimates except for ERA5 and GSMaP, which peak between 23 – 02 LST. In addition, GSMaP and ERA5 show notable differences from the other datasets. In particular, ERA5 has only two different types of clusters: early morning peak hour (green) and early afternoon peak hour (brown). The remaining two clusters in ERA5, late morning peak hour (blue) and afternoon peak hour (red), closely resemble the green and
brown clusters, with only a slight temporal delay.

The spatial distribution of the clusters reveals that afternoon peaks are more frequent over land, while early and late morning peaks occur over the ocean, which is consistent across all datasets (Figure 10). The midnight peak is observed in high latitudes, the oceans of the Southern Hemisphere, as well as over some land regions (i.e., the Great Plains of the USA, northern Africa, the Middle East, parts of north and eastern China, and Australia). Such regional discrepancies are captured and remain consistent
in IMERG, CMORPH, and PERSIANN. In contrast, neither GSMaP nor ERA5 exhibits the midnight peaks. Instead, it is slightly delayed and these regions are occupied by the early and late morning peaks in GSMaP and ERA5. Conversely, the early and late morning peaks in ERA5 are not as pronounced as in GSMaP, especially in Africa. Over the ocean, IMERG and CMOPRH show more or less a similar pattern, with early morning peaks in most regions, late morning peaks in coastal regions, and nocturnal peaks towards high latitudes. For PERSIANN, the nocturnal peaks are more pronounced than for its
counterparts IMERG and CMORPH. As far as GSMaP and ERA5 are concerned, both do not show the diversity observed in the remote sensing datasets, especially in ERA5.

## 4 Discussion

Our findings demonstrate that all the datasets agree in producing the broad spatial pattern and represent the major global precipitation features (e.g., high precipitation ITCZ, SPCZ, and low precipitation dry regions, etc.) across the globe. Nonetheless,
regional inconsistencies do exist among them. These regional disagreements can be observed already through the latitudinal zonal precipitation. Additionally, the precipitation estimates among the products have shown some uncertainties in the dry





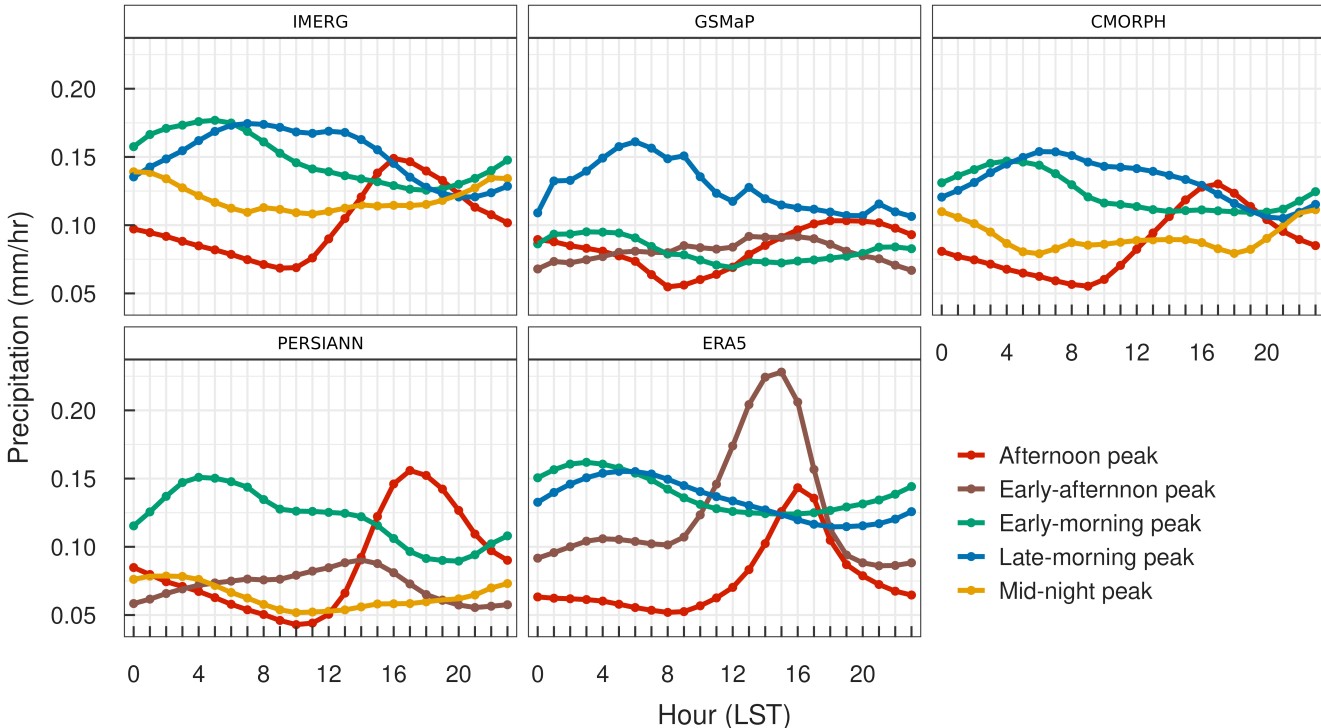

**Figure 9.** K-means clustering (k = 4) produced distinct clusters illustrating the diurnal variation shapes of mean hourly precipitation.

regions such as the Sahara region, northern Africa, Asia, and the low precipitation regions of the Atlantic and Pacific Ocean. Consistent with other studies (Sun et al., 2018; Cattani et al., 2016; Dinku et al., 2011), our findings highlight higher discrepancies among precipitation estimates for drier regions compared to humid regions. The lack of efficient ground observations over these dry and sparsely populated regions could contribute to the large observed uncertainties.

Another important aspect is the high uncertainty among the datasets over the southern hemisphere, especially within the latitudinal range of 30°S – 60°S. Even though similar concerns have previously been reported, in particular over the Southern Ocean (Duque et al., 2023; Siems et al., 2022; Watters and Battaglia, 2021; Behrangi and Song, 2020), the exact reason for such behaviour is yet to be known. A fundamental challenge in this perspective is the lack of long-term, high-quality ground truth over the Southern Ocean(Siems et al., 2022). Furthermore, as reported by Behrangi and Song (2020), the Southern Ocean exhibits the highest precipitation frequency (40%) in terms of zonal averages, and most of the precipitation occurs in the form of light precipitation. As the accurate detection of light precipitation is a persistent problem among the satellite and reanalysis datasets, this could be another probable reason for such discrepancies among them. Additionally, due to fundamental disparities in landmass distribution between the hemispheres, the Southern Ocean experiences distinct influences from atmospheric and oceanic circulation. This results in the formation of unique cloud and precipitation systems, contributing to variations in the intensity and frequency of precipitation when compared to the northern hemisphere (Siems et al., 2022). The large inconsis-




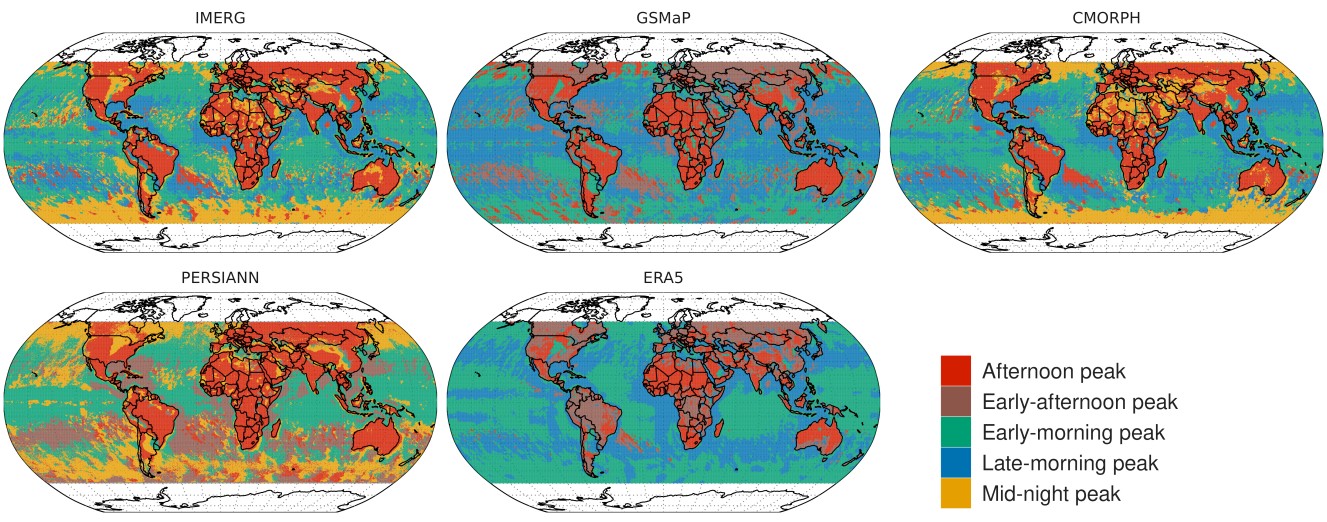

Afternoon peak
Early-afternoon peak
Early-morning peak
Late-morning peak
Mid-night peak

**Figure 10.** Spatial distribution of K-means clustering (k = 4) produced distinct clusters illustrating the diurnal variation shapes of mean hourly precipitation.

tencies among the products over these regions highlight the need for further studies exploring the physical mechanism behind such behaviours.

As previously stated, ERA5, being a reanalysis product, exhibits a slight deviation in behaviour compared to satellite es-
timates, featuring high frequency and low intensity. To further confirm that ERA5's overestimation of frequency is mainly contributed by the light precipitation events, we we conducted our analyses using different thresholds for the definition of wet time intervals (0.2 mm/hr and 0.5 mm/hr). The results (Figure S2) illustrate that as the precipitation threshold increases, the overestimation of the ERA5 frequency gradually decreases and the variation between datasets decreases, confirming the challenge of ERA5 in estimating light precipitation events. Indeed, the 'drizzle problem', characterized by excessively frequent
low-intensity precipitation, is a prevalent pattern consistently observed in model simulations (Dai and Trenberth, 2004). It is mainly attributed to poor representation of convection and model characterizations (Watters et al., 2021). In particular, for ERA5, these high-frequency and low-intensity behaviours have already been reported across the regions, i.e, Eastern China (Qin et al., 2021), Tibetan Plateau (Hu and Yuan, 2021; Chen et al., 2013), eastern Himalaya (Kumar et al., 2021), Alpine Basins over Italy (Shrestha et al., 2023), Africa (Terblanche et al., 2022), southern ocean (Duque et al., 2023), etc., among oth-
ers. Most of the above studies further reported that although ERA5 overestimates the frequency due to its low intensity, it is in better agreement with the total amount of precipitation. In fact, Duque et al. (2023) also found that despite the high-frequency



and low-intensity issues, ERA5 has a better estimate of total precipitation than IMERG over the southern oceanic region. This is probably related to a compensation of the underestimated high intensities (Qin et al., 2021).

Compared to other estimates, GSMaP shows the lowest mean precipitation over land, especially during peak hours between
14:00 and 19:00 LST (Figure 5a). The underestimation of precipitation by GSMaP-MVK compared to the reference datasets is also found in India (-20%) (Prakash et al., 2016). Similar results for GSMaP-MVK V-04 and V05 are also observed over China at the daily scale (Qin et al., 2014; Chen et al., 2015). GSMaP also appears to overestimate frequency, particularly over land (Figure 5a), mainly due to the prevalence of light precipitation events (Figure S2). This is consistent with Qin et al. (2014), who found an overestimation of 20-50% of light precipitation events over China. Despite the overestimation of frequency, the
underestimation of mean precipitation amount and intensity (Figure 5c) suggests that GSMaP either misses the detection of heavy precipitation events or inadequately estimates their magnitude. Compared to CMORPH and IMERG, GSMaP was also found to perform worse against gauge data over Bangladesh (Roy and Banu, 2021). Additionally, the same authors note that GSMaP does not show any precipitation amounts greater than 20 mm, although some regions experience such events.

In contrast, the superior performance of GSMaP-V07 (gauge-corrected) compared to IMERG-F-V06 (underestimation) is
also reported over Luzon compared to the reference datasets (Lee and Huang, 2023). The GSMaP products are also found to be satisfactory in diurnal cycle estimation over the Indonesian Maritime Continents (Ramadhan et al., 2023). On the other hand, Hsu et al. (2021) found that IMERG-E (V06) is better than GSMaP (V07) near-real-time products in estimating the diurnal cycle of precipitation over Taiwan. Given the contradictory findings that vary by region, product version, and evaluation scale (sub-daily/daily/monthly), it is challenging to conclude whether IMERG overestimates the precipitation features or GSMaP
underestimates them.

The early peak hours of the diurnal cycle by ERA5 are also consistently reported by previous studies (Hayden et al., 2023; Chen et al., 2023; Watters et al., 2021). The early peak bias in ERA5 could be mainly attributed to an ineffective representation of the convection in the parameterization schemes (Chen et al., 2023). In particular, the premature convection in the tropics could be the probable reason (Watters et al., 2021). In contrast, the peak of mean precipitation amount in PERSIANN (16 LST)
shows a slight delay compared to the IMERG (15 LST) over land (Figure 5a), which is consistent with the results by Pfeifroth et al. (2016) who reported a 2h delay by PERSIANN over western Africa. They attributed the delay to the infrared-based estimation of precipitation by PERSIANN. The convective clouds in tropical regions primarily precipitate during their early development stages, with less precipitation occurring later when a high, cold ice shield persists. Infrared estimates, relying on cloud top temperature, may tend to overestimate precipitation during the later stages of the convective cloud life cycle.

In addition, PERSIANN tends to overestimate precipitation amounts from the afternoon onwards (15:00 - 21:00 LST), particularly over land, while it remains the lowest over the ocean (Figure 5a). Similar results are also observed by Choumbou et al. (2021) over Central Africa, where they find that PERSIANN overestimates the mean precipitation amount over the Sahel, Cameroon highlands, and Congo basin, but underestimates over the Atlantic Ocean compared to TRMM. In fact, the PERSIANN overestimation of precipitation over Africa can also be observed in Figure S2 (sharp peak between 10°N – 15°N
over land during JJA) and Figure S3. This is consistent with the finding of Pfeifroth et al. (2016) over western Africa, especially over the Niamey mesosite, where PERSIANN estimates more than twice the precipitation amount compared to the reference.





Unlike PERSIANN, CMORPH has overall consistent performance with IMERG, likely due to the incorporation of similar PMW estimates. Moreover, CMORPH also provides a realistic estimation of the diurnal cycle of precipitation in various regions of the world, e.g., China (2018), Africa (Pfeifroth et al., 2016), South America (Giles et al., 2020), North America, and global levels (Janowiak et al., 2005; Tang et al., 2021).

Furthermore, similar to the previous studies, our results also report some unique diurnal regional features across the globe. For example, one such instance is the eastward shift of the diurnal peak over the central United States (Tan et al., 2019). All the estimates depict the nocturnal peaks across the Great Plains, a unique feature of the US diurnal cycle. Nevertheless, unlike the IMERG, GSMaP, CMORPH and, to some extent, PERSIANN, the changes in peak hours by ERA5 exhibit minimal spatial variation (Figure 6). Another example can be seen in the Amazon region, where ERA5 does not exhibit spatial variation in the peak hour of precipitation amount, but instead shows a widespread noon peak hour between 11 – 12 LST (Figure 6). In contrast, the remaining satellite estimates show substantial spatial variation in the peak hour across the Amazon basin, which is especially more pronounced in GSMaP. These results are consistent with those observed by Hayden et al. (2023) and Tai et al. (2021) for ERA5 over the Amazon region. They further attributed that in ERA5, the precipitation pattern was mainly influenced by the solar cycle rather than finer-scale phenomena (e.g., cold pools), which occur at resolutions finer than those of the model.

Moreover, while all the results are presented in comparison with IMERG, it is important to note that this does not imply that IMERG is considered an accurate reference. Although studies have shown that IMERG can be considered as a reference for the evaluation of other diurnal precipitation estimates globally (O and Kirstetter, 2018; Watters et al., 2021; Tang et al., 2021), it is not error-free and still exhibit considerable discrepancies in certain regions (O and Kirstetter, 2018). Nonetheless, the results of O and Kirstetter (2018) are based on IMERG V04, whereas the IMERG version V06 has shown significant improvements over the earlier ones, i.e., IMERG V06 has a very short lag (average +0.59 hour) over the eastern United States (Tan et al., 2019).

## 5  Conclusions

This study compared four state-of-the-art satellite and one reanalysis precipitation data products at the sub-daily scale. In particular, it compared the diurnal variation of precipitation estimates in terms of mean precipitation amount, frequency, and intensity at the global level. The main findings of the analysis are as follows:

– Overall, all products are more consistent in producing spatial patterns of mean hourly precipitation amount than frequency and intensity.

– The discrepancies among the datasets are most pronounced at high latitudes (30°N/S to 60°N/S) compared to the tropical regions (0°N/S to 30°N/S). The agreement among the estimates is higher in the Northern Hemisphere than in the Southern Hemisphere. In particular, the discrepancies are larger at the latitudes between 35°S to 60°S, which could be mainly due to the prevalence of oceans in the Southern Hemisphere.



- ERA5 significantly overestimates the precipitation frequency, and is characterised by very low intensity compared to the rest of the precipitation products. GSMaP also depicts very low precipitation intensity, which is more pronounced over land than the ocean.

- All datasets effectively capture the major diurnal features: an afternoon peak over land and an early morning peak over the ocean. In terms of inter-product comparison, ERA5 detects the peak slightly earlier, around 15 LST over land, compared to the other datasets, which peak at 16 LST. Moreover, ERA5 tends to overestimate the amount of precipitation compared to IMERG and PERSIANN estimates, while CMORPH and GSMaP consistently show lower values.

- In terms of diurnal frequency, ERA5 precipitation frequency is significantly higher than the rest of the estimates, regardless of whether it is over land, over the ocean, or at the global level. However, compared to the land, the high frequencies seem way more dominant over the ocean.

- Different from precipitation mean and frequency, precipitation intensity exhibits a weaker diurnal cycle. ERA5 displays the lowest precipitation intensity among the estimates, while CMORPH exhibits the highest. Surprisingly, GSMaP also shows the lowest intensity among the datasets, even lower than ERA5 over land.

- All the estimates have smaller discrepancies in the peak hour of mean precipitation amount than in the peak hour of frequency and intensity. The highest discrepancies among the datasets of peak hours are observed mainly over the Southern Ocean.

- The K-means clustering results also depict that all the estimates are consistent in reproducing the early-morning peak over the ocean and the afternoon peak over land. Moreover, the IMERG and CMORPH estimates exhibit a high degree of agreement in terms of diurnal shapes, producing similar patterns. However, the remaining products show variations in their diurnal shapes.

Our study comes with certain limitations, that pave the way to future research. The analysis is carried out at the $0.25° \times 0.25°$ resolution, and hence some uncertainties could be associated with the re-gridding, especially for IMERG and GSMaP, which are available at the original resolution of $0.1° \times 0.1°$. Therefore, future studies could consider the IMERG and GSMaP products and evaluate their diurnal variation at their original resolutions. Future studies can also consider the duration of precipitation to gain more comprehensive insights into how the different precipitation durations have distinct diurnal variations and the mechanism behind each precipitation structure. In addition, the diurnal variation of different precipitation intensities will provide further insights.

One potential direction for future investigation involves examining the IMERG V07, which was just published at the time we write, and identifying the effects of major changes brought about by the recent version compared to its predecessor (i.e., V06). In addition, considering the application-oriented importance of near-real-time satellite datasets, such as IMERG-Early, Late runs, and GSMaP near-real-time version (GSMaP-NRT), assessing their capability and identifying uncertainties in their representation of the diurnal cycle can offer additional perspectives, particularly for regions where these datasets could be



potentially applied. Furthermore, a notable limitation of our current analysis is the strict filtering criteria applied, leading to the exclusion of various precipitation datasets available at very high spatial and temporal resolutions but limited to land (e.g., ERA5land). Therefore, forthcoming research endeavours could incorporate these high-resolution products to provide more detailed insights into the uncertainties associated with estimating diurnal precipitation.

Overall, the study provides an overview of the agreements and disagreements among the precipitation products at a sub-daily
scale on a global level, rather than making claims about the superiority of any one product. The results indicate that all the satellite estimates exhibit a high degree of consistency in certain aspects such as reproducing the overall diurnal pattern and peak hour of maximum precipitation throughout the globe. The small regional changes produced, especially by the PERSIANN and CMORPH, indicate that these products can also be adapted to better understand the diurnal cycle of global precipitation. The ERA5 reanalysis estimates, although it produces the diurnal pattern consistent with the remote sensing estimates, show
pronounced regional differences, and thus care should be taken. Moreover, our results assist to identify regions suitable for employing any of the aforementioned products with minimal impact on outcomes, alongside areas necessitating cautious consideration when applying such datasets. Therefore, this study highlights the importance of integration of multiple sources of datasets and caution in relying on individual precipitation products for a comprehensive understanding and accurate analysis of global precipitation dynamics. In addition, this is the first study to present the diurnal cycle on a global scale, using an ensemble
of satellite estimates and reanalysis products covering two decades of data. Moreover, this could be considered as a global scale reference for quantifying uncertainties in the representation of the diurnal precipitation from the global precipitation datasets.

*Code and data availability.* The data compiled herein and the R code for the figures are publicly available at https://czuvpraze-my.sharepoint.com/:f:/g/personal/pradhan_fzp_czu_cz/EtMtFGna-7NCvZg0St2lOPwBGyhYRI93TamgS_6yKBB9Nw?e=bwPu7q.

*Author contributions.* Rajani Kumar Pradhan: Conceptualization, Formal analysis, Investigation, Writing - Original Draft. Yannis Markonis:
Conceptualization, Supervision, Writing - Review  Editing. Francesco Marra: Conceptualization, Supervision, Writing - Review  Editing. Efthymios I. Nikolopoulos: Writing - Review  Editing. Simon Michael Papalexiou: Writing - Review  Editing. Vincenzo Levizzani: Writing - Review  Editing.

*Competing interests.* The authors declare that they have no conflict of interest

*Acknowledgements.* RKP was supported by the Internal Grant Agency (Project no: 2023B0028), Czech University of Life Sciences Prague.
YM and RKP were supported by the project "Investigation of Terrestrial HydrologicAl Cycle Acceleration (ITHACA)" funded by the Czech Science Foundation (Grant 22-33266M)".





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
