# Peer review of "Diurnal Variability of Global Precipitation: Insights from Hourly Satellite and Reanalysis Datasets"

_EGUsphere, 2024_

## Author Comment (AC1)

**Reply to Reviewer comments**

**Reviewer 1**

Summary

The paper addresses the critical importance of accurately estimating global precipitation, especially at the sub-daily scale, where uncertainties are typically higher than daily, monthly, or annual estimates. The study has been conducted on satellite-based and reanalysis products on a global scale. The study specifically investigates the diurnal cycle of precipitation using a comprehensive analysis of five global precipitation products that provide at least hourly resolution data: The analysis focuses on three key parameters of the diurnal variability of precipitation: precipitation amount, frequency, and intensity. The study highlights the need for integrating diverse datasets to minimize uncertainties and ensure a more accurate analysis of global precipitation patterns. Relying on a single precipitation product can lead to misrepresentations due to the inherent discrepancies in the diurnal cycle estimates.

We thank the reviewer for his/her constructive and encouraging comments, and truly appreciate for the time and efforts to review the manuscripts. We paid detailed attention to all the reviewer's comments/suggestions and have responded accordingly.

**Major Comments:**

If the primary goal is a relative comparison among the products (e.g., to understand differences in product outputs without focusing on absolute accuracy), then comparing the products directly to each other is acceptable. However, for a comprehensive evaluation of accuracy and applicability, it is ideal to use a reliable benchmark dataset. This provides valuable insights into the behavior of each product with respect to the intensity and frequency, allowing for a more robust assessment.

We completely agree with the reviewer that the comparing the datasets with a benchmark

would have definitely added further insights into their accuracy and applicability. However, the lack of a global benchmark product at such high spatial and temporal resolution hinder such studies at the global level.

Indeed, we are interested here in relative comparison, and finding the uncertainties among the estimates rather than finding out the best product. The same is also exclusively highlighted in the conclusion section of the study as; P-23 Line 529 'Overall, the study provides an overview of the agreements and disagreements among the precipitation products at a sub-daily scale on a global level, rather than making claims about the superiority of any one product.'

Why did you not use the latest version of the IMERG product (v07) in this study? Using the most recent version is crucial, as updates in IMERG v07 may include improvements in algorithm accuracy, enhanced calibration, and adjustments based on newer gauge corrections. These enhancements can significantly impact the reliability and precision of precipitation estimates, which is particularly important for a robust inter-comparison of precipitation products. I strongly encourage you to consider using IMERG v07 that has been released in 2023, in your analysis to ensure that your findings are based on the most accurate and current data available.

The study was carried out early in 2023, prior to the release of IMERG V07. Due to certain circumstances the manuscript was postponed. Nonetheless, since IMERG V07 is now available, we agree with the reviewer suggestion that ensuring our findings are based on most accurate, we decided to replace IMERG V06 with IMERG V07 in the revised manuscript.

**Minor Comments:**

For PERSIANN, and CMORPH the link where the data have been downloaded are mentioned, please do the same for IMERG and GSMaP:

The reviewer's remarks will be added in the revised manuscript as follows; "The IMERG precipitation datasets are accessed from the NASA Goddard Flight Center (`https://gpm1.gesdisc.eosdis.nasa.gov/data/GPM_L3/`)."

"The GSMaP estimates are obtained from the JAXA G-Portal (`https://earth.jaxa.jp/gpr/`)."

Line 136-139: please add a reference

The following reference will be added in the revised manuscript; "Subsequently, they are further refined to the desired $0.25° \times 0.25°$ resolution through the nearest neighbour interpolation technique (Thiessen, 1911; Berndt and Haberlandt, 2018)".

Line 170-172: This statement is not fully clear to me. Does it mean that in this case, adding more than 4 clusters improved WSS by less than 7%? And based on that the additional clusters did not significantly enhance clustering quality. Selecting k=4, avoids overfitting while preserving meaningful pattern distinctions in the diurnal precipitation data. Is it correct? If yes, I suggest rephrasing your statement to be more clear.

Indeed, the reviewer understanding is correct. Adding more than 4 cluster resulted in reduction of WSS of less than 7%, which indicate the additional clusters did not significantly improve the clustering quality. To clarify this point to the readers, we have rephrased the sentence as follows: "To determine the optimal number of clusters, the process was iterated from K = 1 to K = 15. Ultimately, k = 4 was selected as the appropriate number of clusters because the reduction in the within-cluster sum of squares (WSS) for additional clusters was below 7%."

Line 216-217: Please mention which index you are referring to, here you refer to intensity, am I right? Do you have an idea of what might be the possible causes for the sharp decline

of GSMaP over south hemisphere?

No, we are refereeing to precipitation amount index in that context. There are three paragraphs, each addressing a different index: the first and second paragraphs refers to precipitation amount and frequency, while the third one focuses on intensity. To make this clear we will cite the figure number there.

Regarding the sharp decline of precipitation features for GSMaP, although the exact reason is yet unknown, it is could be mainly caused by the underestimation of heavy precipitation. We have added further discussion attributing potential causes as follows: "In the southern hemisphere, GSMaP has shown a sharp decline from -30°S until -60°S which could be attributed to the underestimation of heavy precipitation events over the ocean. GSMaP's tendency to underestimate both the amount and frequency of heavy precipitation has also been reported in previous studies (Weng et al., 2023; Ning et al., 2017)."

Line 284-288: Is the only possible reason for differences between IMERG and GSMaP the use of different gauge corrections? What about the variations in their underlying algorithms? In short, while both IMERG and GSMaP utilize a similar constellation of satellite estimates, they differ in algorithms, processing techniques, and data sources, all of which could contribute to the observed discrepancies in precipitation estimates.

I'm also interested in learning more about any overlap between the GPCC and CPC networks, as well as an approximate count of gauges used by each. Additionally, since GPCC and CPC likely apply different quality control, calibration, and interpolation methods, this could further contribute to differences in their final precipitation products, even when some gauges are shared.

We agree with the reviewer's comment that the difference between the IMERG and GSMaP is more than just solely attributed to the gauge correction. Indeed, there is some overlap

between the GPCC and CPC stations (Kidd et al., 2017). However, estimating the exact number of overlapping stations is challenging due to variations in data sources, quality control procedures, and the number of gauges, which fluctuate over time. GPCC uses a time-varying collection of data from over 86,000 stations across the globe (Sun et al., 2018; Schneider et al., 2014). It requires at least 10 years of continuous data for the station to be included in the database. In contrast, CPC utilizes approximately 30,000 stations globally and incorporates historical data and additional sources (such as radar and satellite) to perform quality control on raw datasets. The following discussion has been added to the revised manuscript;

"This behavior of GSMaP over land is notable, considering both IMERG and GSMaP use a similar constellation of satellite estimates. Nevertheless, it should also be noted that both datasets use different gauge corrections over land. IMERG applies GPCC corrections on a monthly scale, while GSMaP utilizes CPC corrections on a daily scale. GPCC accesses from a time-varying collection over 86,000 stations (Sun et al., 2018; Schneider et al., 2014), whereas CPC uses data from 30,000 stations over the globe (Xie et al., 2010). Although there is some overlap among the data sources, both datasets (i.e., GPCC and CPC) exhibit differences due to their underlying quality control measures, interpolation techniques, and other factors. Moreover, beyond gauge corrections, differences between IMERG and GSMaP in their precipitation retrieval algorithms, sampling frequency, and other aspects could also contribute to the observed discrepancies. Among these factors, gauge correction is likely a major reason, as the differences are more pronounced over land than over the ocean. Similar performance of GSMaP has also been reported over China (Weng et al., 2023), where it failed to detect precipitation events during the wet season and underestimated both the frequency and magnitude of precipitation extremes."

Line 303-306: It might be interesting for some readers to have more information on some of the possible reasons for these phenomena! For example, the interactions between Sea Surface Temperature and Boundary Layer: Over oceans, cooling at night allows for more

stable conditions near the surface, which can promote the formation of precipitation in the early morning hours. The surface cooling also reduces convective inhibition, allowing for nighttime or early morning convection. You can also mention other possible reasons such as "latitudinal differences in sunlight and heat retention" etc.

The same comment applies to the "coastal regions" and "over land" parts. Figures 6, 7, and 8: These are very good figures; they're highly informative and well-designed."

Thank you for the insightful suggestion. We will add the physical mechanism responsible for those as; "The diurnal variation in precipitation is primarily driven by the difference in diurnal temperature variation between land and ocean, owing to the ocean's higher heat capacity compared to land. In general, the afternoon peak in precipitation over land is mainly attributed to daytime solar heating, which destabilizes the atmosphere and triggers convection (Yang and Smith, 2006). In contrast, cloud-top nighttime radiative cooling and the resulting thermal instability of the atmosphere cause nocturnal or early morning precipitation peaks over the ocean (Yang and Smith, 2006). Coastal regions, which experience precipitation peaks between those of land and ocean, are influenced by land-sea breeze interactions. Additionally, other mechanisms such as regional topography (e.g., mountains and valleys), latitudinal differences in solar heating, and complex local atmospheric circulations, either individually or in combination further complicate and influence the diurnal variation of precipitation in specific regions (Wang et al., 2023; Ruiz-Hernández et al., 2021)."

Page 15, Figure 7: Do you know why GSMaP displays these north-south striping patterns for the peak precipitation frequency hour?"

Generally, those north-south striping patterns occur in all the datasets, but they are more pronounced in GSMaP. The main reason for this remains unclear, but we will look into it in detail and report the appropriate reasons.

**References**

Berndt, C., Haberlandt, U., 2018. Spatial interpolation of climate variables in Northern Germany—Influence of temporal resolution and network density. Journal of Hydrology: Regional Studies 15, 184–202. URL: `https://www.sciencedirect.com/science/article/pii/S2214581817303361`, doi:10.1016/j.ejrh.2018.02.002.

Kidd, C., Becker, A., Huffman, G.J., Muller, C.L., Joe, P., Skofronick-Jackson, G., Kirschbaum, D.B., 2017. So, How Much of the Earth's Surface Is Covered by Rain Gauges? Bulletin of the American Meteorological Society 98, 69–78. URL: `https://journals.ametsoc.org/view/journals/bams/98/1/bams-d-14-00283.1.xml`, doi:10.1175/BAMS-D-14-00283.1. publisher: American Meteorological Society Section: Bulletin of the American Meteorological Society.

Ning, S., Song, F., Udmale, P., Jin, J., Thapa, B.R., Ishidaira, H., 2017. Error Analysis and Evaluation of the Latest GSMap and IMERG Precipitation Products over Eastern China. Advances in Meteorology 2017, e1803492. URL: `https://www.hindawi.com/journals/amete/2017/1803492/`, doi:10.1155/2017/1803492. publisher: Hindawi.

Ruiz-Hernández, J.C., Condom, T., Ribstein, P., Le Moine, N., Espinoza, J.C., Junquas, C., Villacís, M., Vera, A., Muñoz, T., Maisincho, L., Campozano, L., Rabatel, A., Sicart, J.E., 2021. Spatial variability of diurnal to seasonal cycles of precipitation from a high-altitude equatorial Andean valley to the Amazon Basin. Journal of Hydrology: Regional Studies 38, 100924. URL: `https://linkinghub.elsevier.com/retrieve/pii/S2214581821001531`, doi:10.1016/j.ejrh.2021.100924.

Schneider, U., Becker, A., Finger, P., Meyer-Christoffer, A., Ziese, M., Rudolf, B., 2014. GPCC's new land surface precipitation climatology based on quality-controlled in situ data and its role in quantifying the global water cycle. Theoretical and Applied Climatology 115, 15–40. URL: `https://doi.org/10.1007/s00704-013-0860-x`, doi:10.1007/s00704-013-0860-x.

Sun, Q., Miao, C., Duan, Q., Ashouri, H., Sorooshian, S., Hsu, K.L., 2018. A

Review of Global Precipitation Data Sets: Data Sources, Estimation, and Inter-comparisons. Reviews of Geophysics 56, 79–107. URL: `https://onlinelibrary.wiley.com/doi/abs/10.1002/2017RG000574`, doi:10.1002/2017RG000574. _eprint: https://onlinelibrary.wiley.com/doi/pdf/10.1002/2017RG000574.

Thiessen, A.H., 1911. PRECIPITATION AVERAGES FOR LARGE AREAS. Monthly Weather Review 39, 1082–1089. URL: `https://journals.ametsoc.org/view/journals/mwre/39/7/1520-0493_1911_39_1082b_pafla_2_0_co_2.xml`, doi:10.1175/1520-0493(1911)39<1082b:PAFLA>2.0.CO;2. publisher: American Meteorological Society Section: Monthly Weather Review.

Wang, J., Yuan, H., Wang, X., Cui, C., Wang, X., 2023. Impact of Thermally Forced Circulations on the Diurnal Cycle of Summer Precipitation Over the Southeastern Tibetan Plateau. Geophysical Research Letters 50, e2022GL100951. URL: `https://onlinelibrary.wiley.com/doi/abs/10.1029/2022GL100951`, doi:10.1029/2022GL100951. _eprint: https://onlinelibrary.wiley.com/doi/pdf/10.1029/2022GL100951.

Weng, P., Tian, Y., Jiang, Y., Chen, D., Kang, J., 2023. Assessment of GPM IMERG and GSMaP daily precipitation products and their utility in droughts and floods monitoring across Xijiang River Basin. Atmospheric Research 286, 106673. URL: `https://www.sciencedirect.com/science/article/pii/S0169809523000704`, doi:10.1016/j.atmosres.2023.106673.

Xie, P., Chen, M., Shi, W., 2010. CPC unified gauge-based analysis of global daily precipitation (2010 - 90annual_24hydro). URL: `https://ams.confex.com/ams/90annual/techprogram/paper_163676.htm`.

Yang, S., Smith, E.A., 2006. Mechanisms for Diurnal Variability of Global Tropical Rainfall Observed from TRMM. Journal of Climate 19, 5190–5226. URL: `https://journals.ametsoc.org/view/journals/clim/19/20/jcli3883.1.xml`, doi:10.1175/JCLI3883.1. publisher: American Meteorological Society Section: Journal of Climate.

---

## Author Response (AR1)

**Reply to Reviewer comments**

**Reviewer 2**

**Summary:** The manuscript by Pradhan et al. presents a nice overview of the considerable differences between five global hourly precipitation products, which can be considerable, in particular at such a fine temporal scale. It evaluates 4 different satellite products and a reanalysis product, focusing on precipitation amount, frequency and intensity, and investigating the mean annual diurnal cycle of these. By highlighting the significant uncertainties and the inhomogeneity in these, it emphasizes the need to consider and integrate multiple and diverse products, and avoid relying on a single dataset when using and analyzing global precipitation patterns. It also hints, that there is still a lot of work to be done for improving our measurements and process understanding at the subdaily time scale

**General comments and recommendation:**

The manuscript is well written, it has a clear and logic structure, the data and methods used are mostly well described or supported by relevant sources. The manuscript generally features high-quality and interesting figures.

It is a very timely and interesting study, the use and availability of datasets is steadily increasing, with regular refinements being applied and released. Large scale data sets allow us to improve our understanding of the system, on one side of the underlying processes, linkages and feedbacks, and this in turn –on the other side- allows us for describing ungauged or poorly/less well understood areas of the world. Precipitation is key and one of the most challenging variables in the water cycle (and hydrology?), the current intensification of the latter makes the subdaily time scale increase in relevance, in particular in regard of extremes.

It is great to see the large number of products rather meticulously analyzed, I congratulate the authors for this, and for the interesting selection of products. While I did enjoy reading this thorough paper, I would have a few (major) comments and suggestions:

We thank the reviewer for his/her constructive and very detailed comments on the manuscript. We are glad the reviewer finds our work interesting. We generally agree with the reviewer's comments, and happy to implement the suggested changes to the benefit of the manuscript.

1) The authors perform all their analysis by considering the whole year –as a bulk, without breaking their analyses down to the seasonal scale. I wonder why is this so, and if it's not averaging out everything a bit too much? While this is not relevant in the intertropical zone, to my understanding seasonal precipitation patterns significantly vary throughout the year, at least over land, in the rest of the world. Chen et al. 2024 published recently a study presenting a very similar kind of analysis, but focusing on a single product, IMERG V06. They also show a significant seasonal cluster transition, which I think would be of interest (and relevant?) for the readers when looking at the authors' analyses too. I believe the authors should at least comment on this in their work, and clearly justify why the study sticks to the annual scale, and what does this means.

We agree that seasonal analysis can provide valuable insights, especially in regions outside the intertropical zone where precipitation patterns vary significantly throughout the year. The primary reason for focusing on the annual scale rather than the seasonal scale is to provide a broad, more general perspective on the performance of precipitation products at a global level. This approach allows us to identify the overarching trend and pinpoint regions where discrepancies among the dataset are most pronounced, which serve as a foundation for future, more detailed studies.

While we have conducted seasonal analyses as well, including all related figures in the main manuscript would make it overwhelming and challenging to interpret, compare, and discuss in detail. To address this, we have included some of the seasonal analysis results here (please see Figures 1–6). Nonetheless, we are working on a follow-up study that will dive deeper into seasonal dynamics.

Regarding the study by Chen et al. (2024), they focused on a single product (IMERG V06) and analyzed the diurnal cycle without including multi-product comparisons. In

contrast, our study examines five products and presents some hourly intercomparisons before delving into the diurnal variation. Moreover, as the title suggests, their primary objective was clustering the diurnal cycle, whereas our study emphasizes intercomparison among the estimates. In our case, K-means clustering is employed merely as a tool to facilitate an effective intercomparison of the products, as it would otherwise be challenging to present the diurnal variation at each grid. Including seasonal variations for all these products would make the presentation too complex and cumbersome. Therefore, we have chosen to focus on the annual scale in the main manuscript. Nonetheless, we have cited the sudy by Chen et al in various sections of the manuscript as suggested by the reviewers.

[Figure]

Figure 1: Diurnal variation of precipitation amount (mm/hr).

2) In their study, if I understand this correctly, the authors use all products together for estimating the clusters. There might be products "forcing" a reduction of the number of clusters, e.g. ERA5? It would be interesting to see a K-means clustering performed per precipitation product vs. the K-means clustering performed over all products. I will come back to the K-means clustering also in the specific comments, but in general I believe they should spend some more words and figures on how they performed the clustering.

We appreciate the reviewer's insightful question regarding the clustering methodology.

[Figure]

Figure 2: Diurnal variation of precipitation frequency (%).

[Figure]

Figure 3: Diurnal variation of precipitation intensity (mm/hr).

[Figure]

Figure 4: Peak hour of mean precipitation for each dataset during 2001 − 2020.

[Figure]

Figure 5: Peak hour of mean precipitation frequency for each dataset during 2001 − 2020.

[Figure]

Figure 6: Peak hour of mean precipitation intensity for each dataset during 2001 − 2020.

To clarify, the K-means clustering was applied separately to each precipitation product (e.g., IMERG, ERA5) rather than combining all datasets into a single analysis. This approach ensures that the clustering reflects the unique diurnal cycle characteristics of each product. Regarding the concern about ERA5 potentially "forcing" a reduction in cluster numbers: since each product was clustered independently, no single dataset (including ERA5) influenced the clustering of others.

The choice of K=4 was based on systematic testing of cluster numbers (K=4 to K=15). Higher values (e.g., K=6 or K=15) produced fragmented clusters that were difficult to distinguish visually and interpret consistently across products (see attached Figure 7). Restricting the analysis to K=4 balanced clarity with meaningful representation of diurnal cycle phases (e.g., morning, afternoon, evening peaks). To improve clarity, we have revised the methodology section by adding the following details (Line 171-179):

[Figure]

Figure 7: K-means clustering (k = 6) produced distinct clusters illustrating the diurnal variation shapes of mean hourly precipitation.

"The K-means clustering was applied separately for each dataset to ensure that the clustering reflects the unique diurnal cycle characteristics of each product, and no single dataset

influenced the clustering of others. To determine the optimal number of clusters, the process was iterated from K = 1 to K = 15. Ultimately, k = 4 was selected as the appropriate number of clusters because the reduction in the within-cluster sum of squares (WSS) for additional clusters was below 7% (as determined by the Elbow method; see Fig. S7). Another main reason for limiting the number of clusters to four was to ensure distinct, well-separated clusters that are visually differentiable and easily identifiable. Increasing the number of clusters beyond four typically resulted in similar shapes with minimal differences, making them challenging for intercomparison across datasets (see Fig S8). Subsequently, four clusters are extracted from each dataset to depict global diurnal precipitation variability. Finally, each cluster is named according to its peak hour of local solar time."

3) The study might profit by better/more clearly "characterizing" the different products, and in particular indicating if there might be a product, which could be regarded as a sort of benchmark. While I do understand that the main objective of the authors it's rather a relative comparison of the 5 products, and implicitly saying we don't know the truth, we just want to have an insight on the uncertainty, it would be very valuable for a reader, to have a better overview of the accuracy and suitability to use these data, e.g. as input to other models or for other kind of analyses?

We appreciate the reviewer's feedback and completely agree that any insights regarding product accuracy and usability can be greatly beneficial to the scientific community. However, as mentioned, the lack of a reference dataset with such high spatiotemporal resolution at a global level limits the scope of such an analysis. Nonetheless, readers can still benefit from our results, which highlight regions where the products exhibit greater or lesser dependencies. Care should be taken when considering datasets for these regions. For instance, if all datasets show higher discrepancies in a particular region, it would be advisable to prioritize observational data if available. If such data is unavailable, it would be prudent to either consider multiple products or at least acknowledge the discrepancies among the datasets when drawing conclusions. This point is emphasized in our conclusion section as

well (Line 566–567).

4) My last comment it's more a minor comment: I guess it might have to do with the usual delays life brings along, but is there a reason why the authors didn't consider using IMERG V07? The authors write that this dataset was released while writing the paper, but this dataset was released in 2023? Zhu et al. 2024 have recently shown, that there is a clear "hydrologic utility" in using IMERG V07 instead of IMERG V06.

We share the reviewer's concern that now it is more relevant to consider IMERG V07 instead of V06. As we have already addressed a similar concern raised by Reviewer 1, the analysis was initiated in 2023 when V07 was not yet available. Nonetheless, considering the relevance and importance to the scientific community, we have updated all analyses and figures to use IMERG V07 in the revised manuscript. While the differences between V06 and V07 are relatively minor, we ensured consistency by revising all relevant sections. Consequently, the overall results and conclusions remain largely unchanged.

Because of these considerations, I think the manuscript requires some small corrections and some additional work, before it can be recommended for publication. Please find my specific and technical comments here following.

Specific Comments:
Introduction:

1) P2-L31: are there any better – in the sense of more general/less region-specific- references here?.

Yes, we have added the following references (Line 31); (Watters et al., 2021; Dai and Trenberth, 2004).

2) P2-L46-P3-L62: I guess it would help stating somewhere that generally in this paragraph gauge measurements are taken as the ground-truth, and this case what are their limitations

We thank the reviewer for the suggestion. The main idea of this paragraph is to review

studies that used satellite products other than IMERG, since the previous paragraphs focus on IMERG-based studies. The purpose is not to discuss whether they used gauge data or other references, though these regional studies did use gauge data as a reference. We would like to keep it as it is to maintain better flow and coherence. Regarding the limitations of the studies, we have already discussed them in lines 73 – 79.

**Data and Methodology:**

3) I would number the subsections here (e.g. 2.1 Satellite-based datasets,..)?

We agree with the reviewer suggestion, and has been addressed in the revised manuscript.

4) P4-L114: Would PERSIANN PDIR-Now also be option? If not, why?

Yes, PERSIANN PDIR-NoW could be an option. Nonetheless, the main reason behind prioritizing PERSIANN over PERSIANN PDIR NOW is to be consistent with the research-based datasets, such as IMERG-F and GSMaP final research-based products have been selected over their near real-time counterparts. Similarly, as the PERSIANN PDIR-NoW is a near real-time product and the PERSIANN is a research-based one, hence the latter is prioritized over the former.

5) P5-L128: this sentence might be redundant, you already say that the common overlapping period is 2001-2020 shortly after (P5-L133)

The sentence has been removed from the revised manuscript.

6) I would rather improve Table 1: this could be extended to include information on the inclusion of gauge measurement products (which product is used to correct the data, at which temporal scale), and how these are included (algorithms). Readers that are not acquainted with all products would profit from this information, and the expert readers would get a nice and fairer overview of the tiny (but important?) differences between the products analyzed.

We have extended the table by including the datasets they used, and the temporal scale for

Table 1: Summary of the datasets used in this analysis.

| Dataset name | Spatial scale | Temporal scale | Record length | Gauge correction | Reference |
|---|---|---|---|---|---|
| IMERG | $0.1° \times 0.1°$ | 0.5h | 2000 – present | monthly GPCC | Huffman et al. (2015) |
| GSMaP | $0.1° \times 0.1°$ | hourly | 2000 – present | daily CPC | Mega et al. (2019) |
| ERA5 | $0.25° \times 0.25°$ | hourly | 1950 – present | None | Hersbach et al. (2020) |
| PERSIANN | $0.25° \times 0.25°$ | hourly | 2000 – present | None | Hsu et al. (1997) |
| CMORPH | $0.25° \times 0.25°$ | hourly | 1998 – present | daily CPC | Joyce et al. (2004) |

their correction. The correction of rain rates from the satellites is mainly based on optimal theory, i.e., in the case of GSMaP (IMERG), the precipitation rate is adjusted so that the sum of the 24-hour (monthly) GSMaP (IMERG) precipitation approximately matches the CPC (GPCC) gauge measurements (Mega et al., 2014). Providing detailed information about the algorithms and how they integrate the gauge data is not feasible here; instead, we recommend interested readers to see the references provided for such technical details.

7) P5-L137-139: is there a reason why you didn't use a mass-conserving interpolation?

The main reason behind choosing the nearest neighbor method over others is it is one of the simplest and most widely used methods for precipitation (Thiessen, 1911; Berndt and Haberlandt, 2018). However, we agree that the selection of interpolation is subjective, and there is often one could find more than one type of interpolation technique that fits the requirement. Moreover, using the nearest neighbor method ensures that the original values remain unchanged; since our objective is to compare products, rather than having hydrological analyses, our choice leaned toward a method that would not affect our comparison in any way.

8) P6-L170-173: I thing this part on the K-means clustering should be better explained and "back-uped".First, I would provide the plot showing the reduction of WCSS in function of k as Supporting Material. Second, Choosing 7% is somehow a rather subjective choice (if I understand it correctly, in principle you apply the Elbow Method), ideally you would check

also the Silhouette Score? It is interesting to see that Chen et al. 2024 have significantly more clusters (double) when applying clustering on the diurnal cycle on IMERG V06. If I have estimate it correctly, from Figure S1 applying your threshold, this would results in about 6 clusters. Could you please comment on these considerations?

We agree with the reviewer's concerns. The primary reason for choosing four clusters (7%) is that they clearly depict distinct patterns. We acknowledge that the elbow method is generally considered an efficient way to determine the appropriate number of clusters, and indeed, we have applied this approach (Figure 8). However, following the elbow method, adding more than four clusters does not provide additional insights into the shape of the diurnal cycle. In fact, it results in clusters with very minimal differences, making interpretation and intercomparison among the datasets challenging. Nevertheless, as per the reviewer's suggestion, we have included a plot showing the reduction of within-cluster sum of squares (WCSS) as a function of K in the supporting material and added explanatory text to support this decision as follows (Line 171-179);

The K-means clustering was applied separately for each dataset to ensure that the clustering reflects the unique diurnal cycle characteristics of each product, and no single dataset influenced the clustering of others. To determine the optimal number of clusters, the process was iterated from K = 1 to K = 15. Ultimately, k = 4 was selected as the appropriate number of clusters because the reduction in the within-cluster sum of squares (WSS) for additional clusters was below 7% (as determined by the Elbow method; see Fig. S7). Another main reason for limiting the number of clusters to four was to ensure distinct, well-separated clusters that are visually differentiable and easily identifiable. Increasing the number of clusters beyond four typically resulted in similar shapes with minimal differences, making them unsuitable for intercomparison across datasets (see Fig S8). Subsequently, four clusters are extracted from each dataset to depict global diurnal precipitation variability. Finally, each cluster is named according to its peak hour of local solar time.

Regarding the study by Chen et al. (2024), they used the Bisecting K-Means algorithm,

[Figure]

Figure 8: Determination of best cluster using the Elbow method (an example for the CMORPH dataset).

a hierarchical variant of the K-Means algorithm, whereas we employed standard K-Means clustering. Although their study found no significant difference between the two clustering algorithms, they used only a single dataset (IMERG), compared to the five datasets used in our study. If we apply K-Means clustering to each dataset individually, we obtain a different number of clusters for each, making intercomparison among the datasets challenging. To address this, we classified the clusters based on the peak hour of local solar time (e.g., all clusters with peaks between 14–17 LST are labeled as "afternoon peak"). For instance, in Figure 1.a of Chen et al. (2024), Cluster 7 and Cluster 8 have peak hours at 17 and 18 LST, respectively. In our approach, both would be grouped into a single cluster labeled "afternoon peak hour". Despite these visualization preferences, both studies yield similar results, showing afternoon peak hours over land, early morning peaks over the ocean, and intermittent patterns in regions with complex topography and coastal regions.

**Results**

9) P7-L184: here I would rather say that the largest differences are between PERSIANN and all the other products?

We agree with the reviewer's remarks that we can say the largest differences are between PERSIANN and all the other products. The sentence has been rephrased as (Line 189-190): "However, regional differences in the dry regions (e.g., southern Pacific Ocean, Southern Atlantic Ocean, and southern Indian Ocean near Australia) can be observed between the PERSIANN and all other products."

10) P9-L208: ERA5 (0.3 mm/hr) => this values should be larger according to Figure 4a?

We appreciate the reviewer for pointing this out. It is indeed 0.33, and this has been corrected in the revised manuscript (Line 214).

11) P9-L212-213: "This could be related..reanalysis simulations." => besides for PERSIANN?

We have removed the reanalysis simulation (as ERA5's assimilation with the observational dataset is limited to Contiguous USA) from the sentence, as it generally refers to satellite products. The sentence has been rephrased as (Line 219): "This could be related to the lower availability of ground observations that are used to calibrate and adjust satellite products."

12) P9-L221: 10-35 ° rather than 20-30°?

(Line 229): We agree, it is close, so we have replaced it with 10-35°.

13) P11-L266: I would rather argue that in general the relative variation it's stronger over land? Maybe I misunderstood..in general also in the following sentences it's not completely clear to me what the authors exactly mean with amplitude: do you mean spread? The diurnal variation?

Of course, in terms of relative variation, it is stronger over land compared to over the ocean. However, what we mean here is that, compared to other datasets, ERA5 shows a more

prominent diurnal cycle over the ocean with a distinct peak in the early morning, whereas the other datasets exhibit flatter patterns. To clarify our intention, we have slightly revised the sentence as follows (Line 274-276): "However, over the ocean, ERA5 shows a distinct peak in frequency early in the morning (between 03 to 04 LST), with its diurnal variation being more predominant than that of the other datasets.

14) P12-L274-275: This is comparable to the frequency, but here there is the largest spread?

What we mean is that both precipitation amount and frequency exhibit similar diurnal variations, with a distinct peak occurring around the same time (i.e., afternoon, especially over land). However, the intensity shows a different pattern. If we state that the intensity also looks similar to frequency but with a larger spread, it might confuse readers, as they could interpret it as having the same diurnal frequency with a peak in the afternoon. Therefore, we would like to keep it as it is.

15) P13-L284-288: Why would this be a good reason? This is not clear from the text. Consider rewriting the last sentence and in particular better articulate your explanations. There are quite a few papers also dealing with the quality-control and correction factors applied in global precipitation products basing on gauge measurements (Schneider et al 2014; Baudoin et al. 2020, Ehsani et al 2022, etc..I am an hydrologist, so I might be missing the most relevant literature..but still, the authors can definitely do better here

Because the difference between IMERG and GSMaP is more pronounced over land than over the ocean. Both datasets have gauge corrections over land and do not have any such corrections over the ocean. Although we need to compare their near real-time version to further confirmation, based on the results it seems the gauge correction could be a major factor. Besides, there could be other reasons such as retrieval algorithm, data sources, sampling frequency, etc can contribute to the differences among the estimates. GSMaP uses the 'microwave radiometer precipitation retrieval algorithm' over the regions where the microwave radiometer has passed, and the 'microwave radiometer-infrared radiometer combined algorithm' in non-passing areas. Therefore, precipitation retrievals from the

microwave radiometer overpass regions have a higher accuracy than the non-pass regions. In addition, because of the PMW-sensors features, even in the microwave overpass regions, the accuracy is generally better over oceans than over land and this could be another major contribution to the difference observed over land. Moreover, precipitation retrievals challenging conditions such as from coastal regions, orographic precipitation, and snow-covered regions can also contribute to those differences. As Reviewer 1 also raises a similar concern, we have revised and added the following text to enhance clarity (Line 293-304);

"This behavior of GSMaP over land is notable, considering both IMERG and GSMaP use a similar constellation of satellite estimates. Nevertheless, it should also be noted that both datasets use different gauge corrections over land. IMERG applies GPCC corrections on a monthly scale, while GSMaP utilizes CPC corrections on a daily scale. GPCC accesses from a time-varying collection over 86,000 stations (Sun et al., 2018; Schneider et al., 2014), whereas CPC uses data from 30,000 stations over the globe (Xie et al., 2010). Although there is some overlap among the data sources, both datasets (i.e., GPCC and CPC) exhibit differences due to their underlying quality control measures, interpolation techniques, and other factors. Moreover, beyond gauge corrections, differences between IMERG and GSMaP in their precipitation retrieval algorithms, sampling frequency, and other aspects could also contribute to the observed discrepancies. Among these factors, gauge correction is likely a major reason, as the differences are more pronounced over land than over the ocean. Similar performance of GSMaP has also been reported over China (Weng et al., 2023), where it failed to detect precipitation events during the wet season and underestimated both the frequency and magnitude of precipitation extremes."

16) Figures 6-8: It might sound picky and annoying..the figures are great, but aren't there better colors scales?

We agree that the color scales could be improved. We have tested several alternative color schemes; however, the current one performs relatively better in terms of visualization, particularly in distinguishing the different elements. Specifically, while some color palettes worked well for one or two figures, they did not maintain consistency or clarity across all

three figures. For this reason, we would like to retain the current color scale in the revised version.

17) P14-L315-316:"..high degree of similarity.." => over land, but not over ocean?/ ".. noticeable differences.."=> it depends on the continent?

Yes, we agree with the reviewer's concerns. Here, we are mainly discussing about more general overview rather than focusing on continent-specific details. However, considering the reviewer's feedback, we have slightly revised the sentence as follows (Line 340-342): "IMERG and CMORPH show a high degree of similarity, particularly over land, while GSMaP exhibits noticeable differences, which, of course, vary across continents (e.g., the Great Plains of the USA, the Amazon region, and northern Africa).

18) P14-L319: GSMaP and CMORPH almost better agree here than IMERG and CMORPH? E.g. in Africa the late afternoon peak is more present, than in the other two products?

Although GSMaP reflects a similar spatial pattern of peak hours in northern Africa, it clearly shows more prominent nocturnal peak hours in southern Africa, which are not present in either IMERG or CMORPH. Therefore, we would like to keep it as it is.

19) P16-L353: GSMaP and CMORPH not really?

We agree that this is a more generic statement, and some variability exists. To provide more clarity, we have slightly rephrased the sentence as follows (Line 379-380): "Over the ocean, all satellite estimates indicate an early to late morning peak between 03 –– 09 LST, although some heterogeneity exists in CMORPH and GSMaP.

20) Figure 7-8: For GSMaP there seems to be some (satellite related?) artifacts in both frequency and intensity. For intensity also in CMORPH, slightly. Do you happen do know why is it so?

Generally, the north-south striping patterns appear in all datasets and are mainly related to the time conversion from UTC to LST, which we have now corrected. However, some

striping still remains in the GSMaP dataset. We have reviewed this issue but did not find any specific underlying cause.

21) Figure 10: could you please comment on the differences between your analysis and Chen et al. 2024 for IMERG?

The fundamental difference is that Chen et al. (2024) used the Bisecting K-Means algorithm, a hierarchical variant of the K-Means algorithm, whereas we employed standard K-Means clustering. Although their study found no significant difference between the two clustering algorithms, they used only a single dataset (IMERG), compared to the five datasets used in our study. Using more than four clusters results in similar diurnal patterns with minimal differences, which makes intercomparison among the datasets more challenging. To address this, we classified the clusters based on the peak hour of local solar time (e.g., all clusters with peaks between 14–17 LST are labeled as "afternoon peak"). For instance, in Figure 1.a of Chen et al. (2024), Cluster 7 and Cluster 8 have peak hours at 17 and 18 LST, respectively. In our approach, both would be grouped into a single cluster labeled "afternoon peak hour." Despite these methodological differences, both studies yield similar results, showing afternoon peak hours over land, early morning peaks over the ocean, and intermittent patterns in regions with complex topography and coastal regions.

The following has been added in the revised manuscript (Line 418-421:): "A recent by Chen et al. (2024) reported similar results, although they used a hierarchical variant of the K-means algorithm and restricted their analysis to a single dataset, IMERG V06. While their study explored a larger number of clusters, our results are broadly consistent, with both studies identifying afternoon peaks over land, early morning peaks over the ocean, and intermittent patterns in regions with complex topography and along coastal areas."

22) P 19-L421: what do you mean with "model charactererizations"?

We are referring to ERA5 here, specifically its physics and model parameter characterizations. To be more clear to the readers we have rephrased it as follows (Line 450-451); "It is mainly attributed to poor representation of convection and model characterizations (e.g.,

moist convection, planetary boundary layer schemes in atmospheric models) (Watters et al., 2021)."

- Discussion:

23) P21-L480-482: are these the best examples?

According to our knowledge, yes.

- Colnclusions:

24) P23-L533-534: what do you mean with this sentence? ("The small. . . .of global precipitation") Adapted how?

By "small regional changes," we refer to region-specific features such as the nocturnal peaks across the Great Plain, Amazon region, Tibetan Plateau, etc. For better clarity we have slightly revised the sentence as (Line 562-564): "The unique regional features produced (e.g., nocturnal peaks across the Great Plain, Amazon region, Tibetan Plateau, etc.), especially by the PERSIANN and CMORPH, indicate that these products can also be adapted to better understand the diurnal cycle of global precipitation".

Technical corrections:

- P4-L110: GPCP => I believe this acronym was not introduced yet in the text?

Acronym added as (Line 111): "... Global Precipitation Climatology Project (GPCP)" ...

- P6-L171-172: ..reduction in terms of within cluster sum of squares (WCSS)?

The sentence has been rephrased as follows (Line 174-175): "Ultimately, $k = 4$ was selected as the appropriate number of clusters because the reduction in the within-cluster sum of squares (WSS) for additional clusters was below 7% (as determined by the Elbow method; see Fig. S7)."

- P8-L199: ..model reanalyses tend to exhibit..

We are referring to the reanalysis (ERA5). This has been rephrased as (Line 205-206); " This confirms that reanalyses tend to exhibit a high-frequency, low-intensity issue, a concern that has been extensively reported over the years."

- P17-L368: here the colors in brackets are reported wrong and all over the place? I believe grey

should be green, green should be blue and blue should be brown(ish)?

Thanks a lot for pointing this out. This is a typo, and has been corrected in the revised manuscript as follows (Line 394-395); "The clusters are named according to their respective peak hour of local solar time: afternoon peak (red), early morning peak (green), late morning peak (blue), midnight peak (yellow), and early afternoon peak (brown)."

- Figure 9: in the legend correct the typo in Early-afternnon peak =¿ Early-afternoon peak

Corrected.

- P18-L405: add a space before (Siems et al. 2022)

(Line 435) Done.

- P19-L416: .., we we conducted..

(Line 446) Corrected.

- P20-L455: to stay considtent with the rest of the text I would sugest (15:00-21:00 LST)

(Line 486) We have corrected it and ensured consistency throughout the manuscript.

- P21-L464:..e.g., China (2018).. I believe you wanted to cite someone here, it's not clear to me whom..?

(Line 495) Corrected as: "...e.g., China (Chen et al., 2018), ..."

References

Baudouin, J.-P., Herzog, M., and Petrie, C. A (2020).: Cross-validating precipitation datasets in the Indus River basin, Hydrol. Earth Syst. Sci., 24, 427–450, https://doi.org/10.5194/hess-24-427-2020, 2020.

Chen, P., Chen, A.,Yin,S., Li,Y., Liu, J. (2024): Clustering the diurnal cycle of precipitation using global satellite data. Geophysical Research Letters, 51, e2024GL111513, https://doi.org/10.1029/2024GL111513

Ehsani, M. R., Behrangi, A. (2022): A comparison of correction factors for the systematic gauge- measurement errors to improve the global land precipitation estimate, Journal of Hydrology, Volume 610, 2022, 127884, ISSN 0022-1694, https://doi.org/10.1016/j.jhydrol.2022.127884.

Schneider, U., Becker, A., Finger, P. et al. GPCC's new land surface precipitation climatology based on quality-controlled in situ data and its role in quantifying the global water cycle. Theor Appl Climatol 115, 15–40 (2014). https://doi.org/10.1007 s00704-013-0860-x

Zhu, S., Li, Z., Chen, M., Wen, Y., Gao, S., Zhang, J., Wang, J., Nan, Y., Ferraro, S. C., Tsoodle, T. E., Hong, Y., (2024): How has the latest IMERG V07 improved the precipitation estimates and hydrologic utility over CONUS against IMERG V06? Journal of Hydrology, Volume 645, Part B, 2024, 132257, ISSN 0022- 1694, https://doi.org/10.1016/j.jhydrol.2024.132257

**References**

Berndt, C., Haberlandt, U., 2018. Spatial interpolation of climate variables in Northern Germany—Influence of temporal resolution and network density. Journal of Hydrology: Regional Studies 15, 184–202. URL: `https://www.sciencedirect.com/science/article/pii/S2214581817303361`, doi:10.1016/j.ejrh.2018.02.002.

Chen, G., Lan, R., Zeng, W., Pan, H., Li, W., 2018. Diurnal Variations of Rainfall in Surface and Satellite Observations at the Monsoon Coast (South China). Journal of Climate 31, 1703–1724. URL: `https://journals.ametsoc.org/view/journals/clim/31/5/jcli-d-17-0373.1.xml`, doi:10.1175/JCLI-D-17-0373.1. publisher: American Meteorological Society Section: Journal of Climate.

Chen, P., Chen, A., Yin, S., Li, Y., Liu, J., 2024. Clustering the Diurnal Cycle of Precipitation Using Global Satellite Data. Geophysical Research Letters 51, e2024GL111513. URL: `https://onlinelibrary.wiley.com/doi/abs/10.1029/2024GL111513`, doi:10.1029/2024GL111513. _eprint: https://onlinelibrary.wiley.com/doi/pdf/10.1029/2024GL111513.

Dai, A., Trenberth, K.E., 2004. The Diurnal Cycle and Its Depiction in the Community Climate System Model. Journal of Climate 17, 930–951. URL: `https://journals.ametsoc.org/view/journals/clim/17/5/1520-0442_2004_017_0930_tdcaid_2.0.co_2.xml`, doi:10.1175/1520-0442(2004)017<0930:TDCAID>2.0.CO;2. publisher: American Meteorological Society Section: Journal of Climate.

Hersbach, H., Bell, B., Berrisford, P., Hirahara, S., Horányi, A., Muñoz-Sabater, J., Nicolas, J., Peubey, C., Radu, R., Schepers, D., Simmons, A., Soci, C., Abdalla, S., Abellan, X., Balsamo, G., Bechtold, P., Biavati, G., Bidlot, J., Bonavita, M., De Chiara, G., Dahlgren, P., Dee, D., Diamantakis, M., Dragani, R., Flemming, J., Forbes, R., Fuentes, M., Geer, A., Haimberger, L., Healy, S., Hogan, R.J., Hólm, E., Janisková, M., Keeley, S., Laloyaux, P., Lopez, P., Lupu, C., Radnoti, G., de Rosnay, P., Rozum, I., Vamborg, F., Villaume, S., Thépaut, J.N., 2020. The ERA5 global reanalysis. Quarterly Journal of the Royal Meteorological Society 146, 1999–2049. URL: `https://onlinelibrary.wiley.com/doi/abs/10.1002/qj.3803`, doi:10.1002/qj.3803. _eprint: https://onlinelibrary.wiley.com/doi/pdf/10.1002/qj.3803.

Hsu, K.l., Gao, X., Sorooshian, S., Gupta, H.V., 1997. Precipitation Estimation from Remotely Sensed Information Using Artificial Neural Networks. Journal of Applied Meteorology and Climatology 36, 1176–1190. URL: `https://journals.ametsoc.org/view/journals/apme/36/9/1520-0450_1997_036_1176_pefrsi_2.0.co_2.xml`, doi:10.1175/1520-0450(1997)036<1176:PEFRSI>2.0.CO;2. publisher: American Meteorological Society Section: Journal of Applied Meteorology and Climatology.

Huffman, G.J., Bolvin, D.T., Braithwaite, D., Hsu, K., Joyce, R., Kidd, C., Nelkin, E.J., Xie, P., 2015. NASA Global Precipitation Measurement (GPM) Integrated

Multi-satellitE Retrievals for GPM (IMERG). Algorithm theoretical basis document (ATBD) version 4. URL: `https://gpm.nasa.gov/sites/default/files/document_files/IMERG_ATBD_V4.5.pdf`.

Joyce, R.J., Janowiak, J.E., Arkin, P.A., Xie, P., 2004. CMORPH: A Method that Produces Global Precipitation Estimates from Passive Microwave and Infrared Data at High Spatial and Temporal Resolution. Journal of Hydrometeorology 5, 487–503. URL: `https://journals.ametsoc.org/view/journals/hydr/5/3/1525-7541_2004_005_0487_camtpg_2_0_co_2.xml`, doi:`10.1175/1525-7541(2004)005<0487:CAMTPG>2.0.CO;2`. publisher: American Meteorological Society Section: Journal of Hydrometeorology.

Mega, T., Ushio, T., Kubota, T., Kachi, M., Aonashi, K., Shige, S., 2014. Gauge adjusted global satellite mapping of precipitation (gsmap_gauge), in: 2014 XXXIth URSI General Assembly and Scientific Symposium (URSI GASS), IEEE. pp. 1–4.

Mega, T., Ushio, T., Takahiro, M., Kubota, T., Kachi, M., Oki, R., 2019. Gauge-Adjusted Global Satellite Mapping of Precipitation. IEEE Transactions on Geoscience and Remote Sensing 57, 1928–1935. URL: `https://ieeexplore.ieee.org/abstract/document/8485422`, doi:`10.1109/TGRS.2018.2870199`. conference Name: IEEE Transactions on Geoscience and Remote Sensing.

Schneider, U., Becker, A., Finger, P., Meyer-Christoffer, A., Ziese, M., Rudolf, B., 2014. GPCC's new land surface precipitation climatology based on quality-controlled in situ data and its role in quantifying the global water cycle. Theoretical and Applied Climatology 115, 15–40. URL: `https://doi.org/10.1007/s00704-013-0860-x`, doi:`10.1007/s00704-013-0860-x`.

Sun, Q., Miao, C., Duan, Q., Ashouri, H., Sorooshian, S., Hsu, K.L., 2018. A Review of Global Precipitation Data Sets: Data Sources, Estimation, and Intercomparisons. Reviews of Geophysics 56, 79–107. URL: `https://onlinelibrary.wiley.com/doi/abs/10.1002/2017RG000574`, doi:`10.1002/2017RG000574`. _eprint: https://onlinelibrary.wiley.com/doi/pdf/10.1002/2017RG000574.

Thiessen, A.H., 1911. PRECIPITATION AVERAGES FOR LARGE AREAS. Monthly Weather Review 39, 1082–1089. URL: `https://journals.ametsoc.org/view/journals/mwre/39/7/1520-0493_1911_39_1082b_pafla_2_0_co_2.xml`, doi:`10.1175/1520-0493(1911)39<1082b:PAFLA>2.0.CO;2`. publisher: American Meteorological Society Section: Monthly Weather Review.

Watters, D., Battaglia, A., Allan, R., 2021. The Diurnal Cycle of Precipitation According to Multiple Decades of Global Satellite Observations, Three CMIP6 Models, and the ECMWF Reanalysis. Journal of Climate doi:`10.1175/JCLI-D-20-0966.1`.

Weng, P., Tian, Y., Jiang, Y., Chen, D., Kang, J., 2023. Assessment of GPM IMERG and GSMaP daily precipitation products and their utility in droughts and floods monitoring across Xijiang River Basin. Atmospheric Research 286, 106673. URL: `https://www.sciencedirect.com/science/article/pii/S0169809523000704`, doi:`10.1016/j.atmosres.2023.106673`.

Xie, P., Chen, M., Shi, W., 2010. CPC unified gauge-based analysis of global daily precipitation (2010 - 90annual_24hydro). URL: `https://ams.confex.com/ams/90annual/techprogram/paper_163676.htm`.